# BICD2 phosphorylation regulates dynein function and centrosome separation in G2 and M

Núria Gallisà-Suñé [1], Paula Sànchez-Fernàndez-de-Landa[1,4], Fabian Zimmermann[2], Marina Serna[3], Laura Regué[1], Joel Paz[2], Oscar Llorca [3], Jens Lüders [2] & Joan Roig [1]✉

The activity of dynein is regulated by a number of adaptors that mediate its interaction with dynactin, effectively activating the motor complex while also connecting it to different cargos. The regulation of adaptors is consequently central to dynein physiology but remains largely unexplored. We now describe that one of the best-known dynein adaptors, BICD2, is effectively activated through phosphorylation. In G2, phosphorylation of BICD2 by CDK1 promotes its interaction with PLK1. In turn, PLK1 phosphorylation of a single residue in the N-terminus of BICD2 results in a structural change that facilitates the interaction with dynein and dynactin, allowing the formation of active motor complexes. Moreover, modified BICD2 preferentially interacts with the nucleoporin RanBP2 once RanBP2 has been phosphorylated by CDK1. BICD2 phosphorylation is central for dynein recruitment to the nuclear envelope, centrosome tethering to the nucleus and centrosome separation in the G2 and M phases of the cell cycle. This work reveals adaptor activation through phosphorylation as crucial for the spatiotemporal regulation of dynein activity.

Cytoplasmic dynein-1 (herein referred to simply as "dynein") is a highly conserved microtubule motor that has a wide array of functions both in dividing and non-dividing cells[1,2]. As the primary minus-end-directed motor in animal cells, dynein participates in the transport of different types of vesicles, proteins and mRNA particles. In addition, as a result of its ability to exert force on cellular structures, dynein contributes to the positioning of organelles such as centrosomes, the Golgi apparatus and the nucleus[3–5] and has an important role during mitotic spindle formation[6].

Dynein mobility, processivity and ability to exert force are greatly increased by its interaction with its cofactor dynactin plus one of different adaptors. The dynein adaptors are non-related coiled-coil rich proteins that facilitate the interaction between one or two dynein complexes with dynactin, effectively acting as activators of the motor complex. Adaptors do so by providing a ~300 residue long region that is positioned in an extended conformation along the dynactin filament while also interacting with dynein, thus organizing the motor complex in a manner that favors mobility and processivity. Dynein adaptors are additionally in charge of connecting the motor complex with different cellular cargos, resulting in their transport along microtubules[1,7].

One of the most studied dynein adaptors is BICD2. It belongs to the BICD family of coiled-coil proteins, that in humans contains 4 members, BICD1, BICD2, BICDL1/BICDR1 and BICDL2/BICDR2[8]. BICD proteins function as dimers and have all been described as dynein adaptors in different cellular contexts. They are able to bind dynein and dynactin through their N-terminus, while interacting with different

[1]Department of Cells and Tissues, Molecular Biology Institute of Barcelona (IBMB-CSIC), Baldiri i Reixac 10-12, 08028 Barcelona, Spain. [2]Mechanisms of Disease Programme, Institute for Research in Biomedicine (IRB Barcelona), The Barcelona Institute of Science and Technology, Baldiri Reixac 10-12, 08028 Barcelona, Spain. [3]Structural Biology Programme, Spanish National Cancer Research Centre (CNIO), Melchor Fernández Almagro 3, E-28029 Madrid, Spain. [4]Present address: Aging and Metabolism Programme, IRB Barcelona, Barcelona, Spain. ✉e-mail: jrabmc@ibmb.csic.es

cargos through their C-terminus. BICD2 is a ~94 kDa protein that, similarly to the closely related BICD1, is composed of three coiled-coil regions (named CC1-CC3) separated by two flexible linkers predicted to be unstructured. BICD2 interacts with dynein and dynactin through its N-terminal CC1 region and part of the central CC2 region[9,10]. This segment has a rod-shaped structure and is sandwiched between dynein and dynactin[11]. The CC1 contains a motif aptly named CC1 box, which is conserved across the BICD family, and that is crucial for the interaction with dynein[12]. Additionally the CC2 contains a so-called Spindly motif, which interacts with dynactin[13].

The N-terminus of BICD2 is able to induce the formation of processive dynein/dynactin complexes[9,10,14–16]. Importantly, BICD2 CC3 interacts with the CC1 region, reducing the dynein/dynactin-binding activity of the N-terminus, thus effectively acting as an auto-inhibitory domain[9,14]. The CC3 region also interacts with Rab6, a small GTPase involved in the control of membrane traffic, that links the adaptor (and through it dynein) to the Golgi apparatus and exocytotic vesicles[17,18]. The current model for the regulation of BICD2 (and possibly other BICD family members) suggests that binding of cargo such as Rab6 to the C-terminus releases the N-terminal region from its embrace, "opening" the molecule and favoring the formation of active complexes with dynein and dynactin[8,14,17,19]. Rab6 is thought to be the major interacting cargo of BICD2 during G1 and S and in non-dividing cells. In early G2, however, BICD2 switches cargos and binds the nucleoporin RanBP2/NUP358, resulting in its accumulation at the nuclear envelope where it recruits dynein and dynactin[20]. RanBP2 phosphorylation by the cyclin-dependent kinase CDK1 favors its interaction with BICD2, although possibly other regulatory inputs exist[21]. Additionally, other nuclear proteins such as Nesprin-2 collaborate in recruiting BICD2/dynein to the nuclear membrane[22].

The accumulation of dynein at the nuclear envelope plays a major role in tethering the centrosomes and the nucleus, contributing to early centrosome separation and proper spindle assembly[20,23–27], as well as to nuclear envelope breakdown[28]. Interestingly, in neural progenitors it also drives apical nuclear migration, a process mediated in part by BICD2 that is critical for normal division in these cells[21,29,30]. Here we present a new regulatory mechanism controlling dynein recruitment to the nuclear envelope that is based on the sequential phosphorylation of BICD2 by CDK1 and the Polo-like kinase PLK1. Our results show that BICD2 interacts with PLK1 through the kinase polo-box domain (PBD), and that this interaction depends on the phosphorylation of BICD2 by CDK1. In turn, PLK1 is able to phosphorylate Ser102 in the N-terminus of BICD2. We show that modification of Ser102 interferes with the interaction between BICD2's N- and C-terminal regions, likely altering its architecture. Importantly, Ser102 modification increases the ability of BICD2 to bind and activate dynein, and to interact with RanBP2 as well. We highlight the physiological relevance of our observations by showing that phosphorylation of BICD2 is not only necessary for dynein recruitment to the nuclear envelope and proper nucleus-centrosome tethering in G2, but also for subsequent centrosome separation in early mitosis. In summary, we identify adaptor phosphorylation as a novel mechanism to regulate dynein activation and localization.

## Results

### PLK1 PBD interacts with BICD2 in a CDK phosphorylation-dependent manner

Several high throughput studies indicate that BICD2 is phosphorylated in vivo at multiple residues (see https://www.phosphosite.org/ and studies cited therein). We wondered whether some of these post-translational modifications could be modulating BICD2 function, and sought to identify the kinase responsible for them. A good candidate was PLK1, a protein kinase active during G2 and M and with multiple functions during these phases of the cell cycle[31,32]. PLK1 had been previously suggested to interact with BICD2 by various assays including yeast two hybrid[33], affinity purification coupled to mass spectrometry (MS)[34] and proximity labeling-MS[35]. Moreover, some of the BICD2 phosphosites found in vivo conform to a putative PLK1-phosphorylation motif (see below). We thus focused our efforts on this protein kinase. To further confirm that the two proteins could indeed form a complex in mammalian cells, we immunoprecipitated endogenous BICD2 from exponentially-growing and mitotic HeLa cells and probed the precipitates with anti-PLK1 antibodies (Fig. 1A). Indeed, PLK1 was specifically detected in BICD2 immunoprecipitates and the detected interaction was substantially increased in mitosis, when PLK1 is active. Interestingly in this phase of the cell cycle BICD2 showed a higher apparent molecular weight after electrophoresis (see also Fig. 1C), thus suggesting that it was phosphorylated. Additional immunoprecipitations using recombinant fragments of BICD2 indicated that PLK1 preferentially interacted with the region containing the first two coiled-coil regions of the adaptor (BICD2 [1–575]; note that unless otherwise indicated, BICD2 residue numbers in this work refer to the mouse sequence) (Fig. 1B).

PLK1 interacts with its substrates in a phosphospecific manner through its polo-box domain (PBD)[36]. We thus expressed GST-PLK1 PBD (GST-PLK1 [354–603]) in bacteria and determined whether it was able to interact with BICD2 from cell extracts. We indeed observed that GST-PLK1 PBD specifically interacted with BICD2 (Fig. 1C). Moreover, and in agreement with the results obtained with full-length PLK1, the amount of BICD2 associated with PLK1 PBD was significantly higher when mitotic extracts were used. Additionally, we noted the PBD almost exclusively interacted with the form of BICD2 that showed a high apparent molecular weight, prevalent in mitotic cells. Altogether our results indicate that PLK1 interacts through its PBD domain with modified BICD2, through a region in the N-terminal part of the adaptor. This is in agreement with two-hybrid results with *Drosophila* Polo and BICD[33].

The PBD is a phosphoserine/phosphothreonine binding domain that depends on a priming phosphorylation step for its interaction with other proteins, frequently carried out by proline-directed kinases such as the cyclin-dependent kinases (CDKs)[36]. Our results suggest that BICD2 may preferentially interact with PLK1 in G2 and M and thus we inquired whether CDK1, in charge of controlling progression through these phases of the cell cycle, could modify BICD2. For this, we sought to express three different fragments of BICD2 in bacteria, each containing one of the coiled-coil regions of the adaptor. We succeeded in purifying as GST-fusions the two most N-terminal fragments (GST-BICD2 [1–271] and GST-BICD2 [272–540]), comprising most of the region shown to bind PLK1 in Fig. 1B. Upon incubation with CDK1/cyclin B plus ATP/Mg$^{2+}$, GST-BICD2 [272–540] became readily phosphorylated, in contrast to GST-BICD2 [1–271] that was only slightly modified (Fig. 2A, B). Proteolytic digestion followed by liquid chromatography-tandem MS (LC–MS/MS) analysis identified two phosphorylated peptides in GST-BICD2 [272–540]. One of the modifications was unequivocally assigned to a single residue, Ser331, while the other could not be assigned among three contiguous residues ($_{319}$TST$_{321}$) (Fig. 2C, I). All of the CDK1 putative sites cluster in close proximity and have been previously shown to be phosphorylated in humans in vivo (see https://www.phosphosite.org/). Mutation of Thr319, Ser320 and Thr321 to non-phosphorylatable alanines (GST-BICD2 [272-540; T319A, S320A, T321A]) strongly impaired phosphorylation by CDK1/cyclin B, suggesting that at least one of the three residues is the major target of the kinase in the N-terminal region of BICD2 (Supplementary Fig. S1). Thr319, Ser320 and Thr321 and the surrounding residues are mostly conserved in mammals and in other vertebrates (Fig. 2I). Importantly, they contain a canonical CDK phosphorylation motif ([S/T]PX[K/R], where X is any amino acid; see for example[37]) and, when modified, conform to a PBD-interacting motif (S[pS/pT]PX, where pS and pT are phosphoserine and phosphothreonine, respectively)[36].

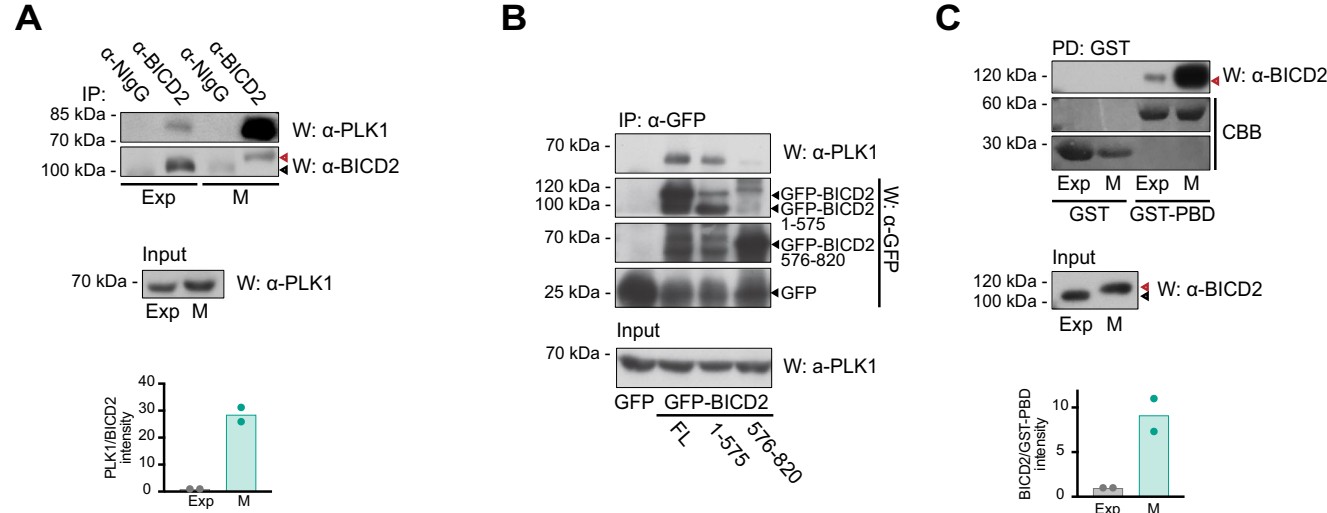

**Fig. 1 | BICD2 interacts with PLK1 through its Polo-Box Domain.**
**A** Coimmunoprecipitation of endogenous PLK1 and BICD2 in exponentially grow-ing (*Exp*) and mitotic (*M*) HeLa cell extracts. anti-BICD2 or normal IgG (*NIgG*) immunoprecipitates (*IP*) were analyzed by western blot (*W*) using either anti-PLK1 or anti-BICD2 antibodies. Note BICD2 high apparent molecular weight in mitotic cells, red arrowheads. PLK1 in the corresponding extracts is shown in the lower panel. The mean of quantifications corresponding to two independent experiments is shown (PLK1 intensity/BICD2 intensity in the immunoprecipitates; this and sub-sequent experiments source data are provided in the Source Data file). **B** PLK1 interacts with the N-terminal region of BICD2. Immunoprecipitates of the indicated recombinant GFP-fusion proteins were analyzed by western blot using either anti-PLK1, to detect the endogenous kinase, or anti-GFP. PLK1 in the corresponding extracts is shown in the lower panel. Note that different fragments of GFP-BICD2 are produced when recombinant proteins are expressed in cells. Arrowheads

indicating the expected MW of relevant proteins have been added for clarity (GFP-BICD2, ~120 kDa; GFP-BICD2 1-575, ~90 kDa; GFP-BICD2 576-820, ~55 kDa; GFP, ~25 kDa). One of two independent experiments with similar results is shown. **C** PLK1 interacts with BICD2 through its Polo-Box Domain (PBD). Extracts of exponentially growing (*Exp*) or mitotic (*M*) HeLa cells were incubated with either bacterial expressed GST or GST-PBD (GST-PLK1 [354–603]) bound to GSH agarose beads. Endogenous BICD2 was detected by western blot (*W*) and GST-fusion proteins by Coomassie Brilliant Blue staining (*CBB*). BICD2 in the corresponding extracts is shown in the lower panel. Note that in both pulldowns from exponential and mitotic extracts the apparent molecular weight of BICD2 interacting with PLK1-PBD corresponds to the fastest migrating form (red arrowhead, high MW; black arrowhead, low MW). The mean of quantifications corresponding to two inde-pendent experiments is shown (BICD2 intensity/GST-PBD intensity in the pulldowns).

To test the hypothesis that CDK1 phosphorylation could be reg-ulating PLK1 binding to BICD2, we next expressed both wild-type BICD2 and a form of BICD2 that cannot be phosphorylated at $_{319}$TST$_{321}$ (BICD2 [T319A, S320A, T321A], henceforth BICD2 AAA) in mammalian cells and determined whether mutation of these residues would interfere with PLK1 binding to BICD2. Indeed, we observed that the GFP-BICD2 AAA mutant did not interact with PLK1 in neither exponentially-growing or mitotic cells (Fig. 2D). Moreover GFP-BICD2 AAA was not pulled down from cell extracts by GST-PLK1 PBD, in contrast to its wild-type counterpart (Fig. 2E). We interpret our data as a confirmation that CDK1 phosphorylation of one or more residues in BICD2 $_{319}$TST$_{321}$ results in the interaction of BICD2 with PLK1 through the PLK1 PBD.

### PLK1 phosphorylates Ser102 in the N-terminal dynein-interact-ing region of BICD2
We next sought to determine whether BICD2 is a PLK1 substrate. For this we incubated GST-BICD2 [1-271] and GST-BICD2 [272-540] with PLK1 plus ATP/Mg$^{2+}$ (Fig. 2F, G). Both polypeptides were phosphory-lated by PLK1, although GST-BICD2 [1–271] was a better substrate, incorporating rapidly up to three times more phosphate than GST-BICD2 [272–540] upon incubation with PLK1. MS analysis identified three BICD2 peptides phosphorylated by PLK1. Two of the modifica-tions were unequivocally assigned to Ser102 and Thr289, while the third could not be assigned among two nearby residues, Thr531 and Ser533 (Fig. 2H, I). Of the identified sites, only Ser102 was located in GST-BICD2 [1–271] suggesting that the modification of this residue accounted for most of the phosphate incorporated in the adaptor in vitro. BICD2 Ser102 is conserved in jawed vertebrates (*Gnathosto-mata*), including tetrapods as well as bony and cartilaginous fish (Fig. 2I), and is surrounded by a sequence that conforms to a canonical

PLK1 phosphorylation site ([D/E]X[S/T]Φ, where X is any amino acid and Φ is an hydrophobic residue[38,39]). The residue and surrounding aminoacids are conserved as well in BICD1 (but not in BICL1 or BICL2, see Supplementary Fig. S2 and the "Discussion"). Additionally, among the phosphosites identified, Ser102 is the only one that has been detected in vivo in human cells. We thus focused our study on the functional role of the phosphorylation of this BICD2 residue.

### Modification of BICD2 Ser102 favors dynein binding and activation
Ser102 is located in the N-terminal region of BICD2, close to the dynein-interacting CC1 box (see Fig. 2I). We reasoned that phosphor-ylation could regulate the ability of this region to form a complex with the dynein motor complex. To investigate this, we mutated Ser102 to non-phosphorylatable (BICD2 S102A) and phosphomimetic (BICD2 S102D) residues. We then expressed the different recombinant BICD2 forms in mammalian cells and observed the amount of dynein and dynactin (detected using anti-dynein intermediate chain (DIC) and anti-dynactin p150$^{Glued}$ subunit (p150) antibodies, respectively) immu-noprecipitating with them. Levels of dynein and dynactin in wild-type GFP-BICD2 and GFP-BICD2 S102A immunoprecipitates were low and barely detectable in some experiments. In contrast GFP-BICD2 [1–575], lacking the cargo-binding regulatory C-terminal region, readily pre-cipitated both dynein and dynactin (Fig. 3A, C). This is consistent with the suggestion that full-length BICD2 exists in a conformation that precludes it from binding dynein/dynactin[9,10]. Importantly, while GFP-BICD2 S102A coimmunoprecipitated amounts of dynein and dynactin than were similar or often lower to those of the wild-type form, the phosphomimetic BICD2 S102D consistently precipitated amounts of both DIC and p150 that were comparable to those associated with BICD2 [1–575]. Thus, the modification of Ser102 favors the interaction

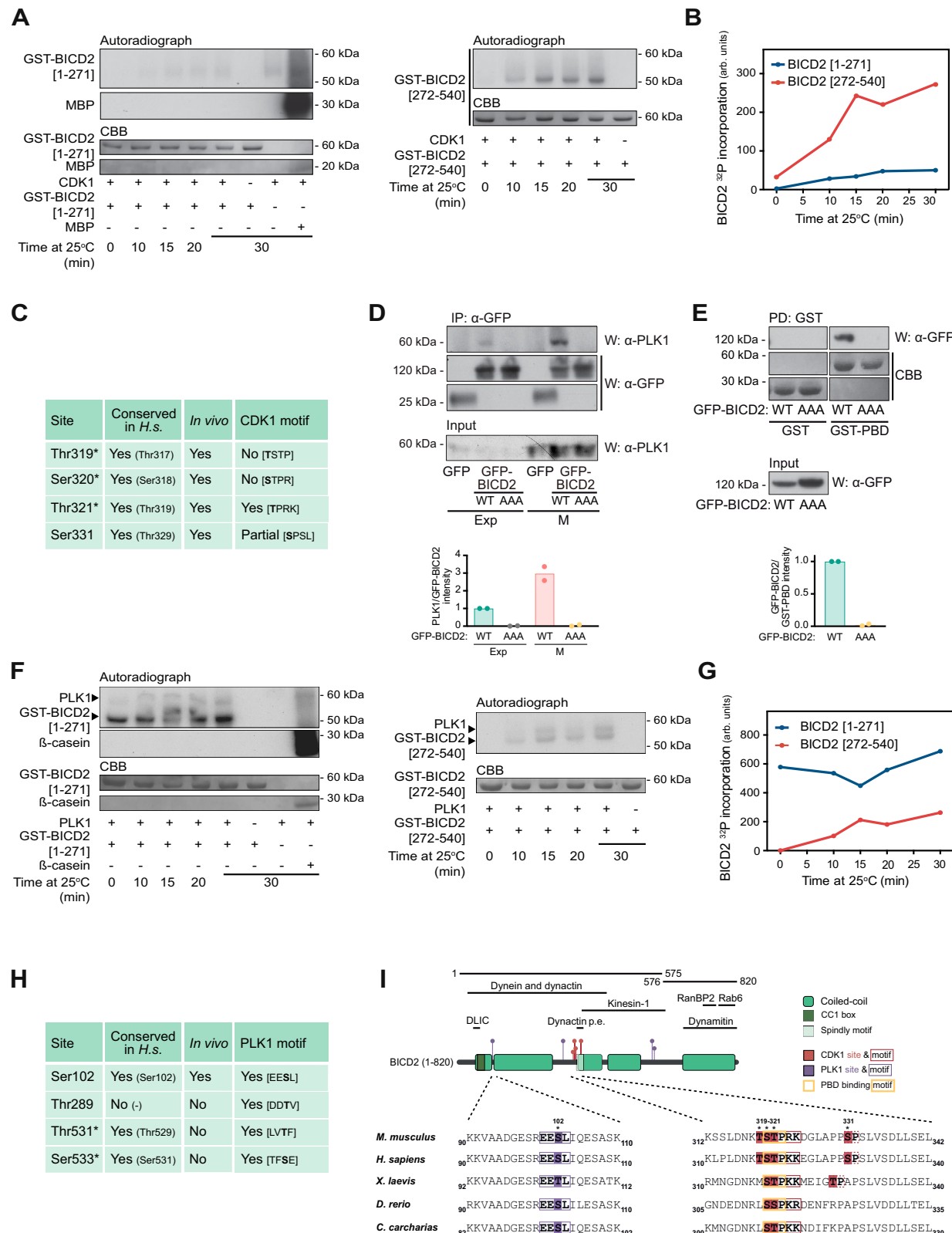

of full-length BICD2 with dynein and dynactin. This was confirmed by experiments in which DIC was immunoprecipitated from cells expressing different forms of recombinant BICD2 (Fig. 3B, D). In DIC immunoprecipitates, wild type and S102A GFP-BICD2 forms were below detection limits, whereas GFP-BICD2 S102D was readily detected. p150 was detected in all cases, possibly resulting from the presence in the cells of dynein/dynactin complexes formed with other adaptors, independently of BICD2.

The interaction of dynein and dynactin with BICD2 has been shown to lead to the formation of a highly processive minus-end-directed motor complex[15,16]. Therefore, we tested whether the modification of BICD2 Ser102 increased dynein mobility in vivo. For this, we

**Fig. 2 | BICD2 is a PLK1 and CDK1 substrate. A** and **B** In vitro phosphorylation of GST-BICD2 fragments by CDK1/cyclin B. Purified bacterial GST-BICD2 [1–271] (left) or GST-BICD2 [272–540] (right) were incubated with recombinant CDK1/cyclin B plus [γ-$^{32}$P]ATP/Mg$^{2+}$ at 25 °C for the indicated times. Myelin basic protein (*MBP*) was used as a positive control. Reactions were stopped with electrophoresis sample buffer and after SDS–PAGE proteins were visualized by Coomassie Brilliant Blue staining (*CBB*) and $^{32}$P incorporation by autoradiography. Relative $^{32}$P incorporation into BICD2 was quantified using a Phosphorimager and is shown in B. A non-radioactive replicate of the GST-BICD2 [272–540] reaction at 15 min was used to identify phosphorylated sites by LC–MS/MS. The original analysis using trypsin covered >80% of the sequence and included all the putative CDK1 sites except Thr321, predicted to be contained in a very short peptide (Thr319-Lys324). The analysis identified Ser331 as phosphorylated site. An additional analysis targeted to Thr321 and its surroundings and using chymotrypsin was carried out, detecting an additional phosphopeptide with a single phosphosite ambiguous between Thr319, Ser320 and Thr321. **C** Sites phosphorylated in vitro in BICD2 [272–540] by CDK1, as identified by LC–MS/MS. *Conserved in H.s.*, conservation in humans (the orthologous residue is indicated between parenthesis). In vivo, sites previously determined to be phosphorylated in vivo in human cells (see https://www.phosphosite.org/). *CDK1 motif*, presence of a CDK1 consensus motif around the phosphorylation site ([S/T]PX[K/R] where X is any residue and S or T are the modified residues). Asterisks denote that a single phosphorylation site was detected that could not be assigned unequivocally among the three residues. **D**, **E** The interaction of BICD2 and PLK1 depends on the residues phosphorylated by CDK1. **D** anti-GFP immunoprecipitates (*IP*) from extracts of exponentially growing (*Exp*) or mitotic (*M*) HeLa cells expressing either GFP, GFP-BICD2 wild type (*WT*) or GFP-BICD2 [T319A, S320A, T321A] (*GFP-BICD2 AAA*). Endogenous PLK1 and GFP are detected by western blot (*W*). PLK1 in the corresponding extracts is shown in the lower panel. **E** Extracts of HeLa cells expressing GFP-BICD2 wild type (*WT*) or GFP-BICD2 AAA were incubated with GST or GST-PBD bound to GSH agarose beads. GFP was detected by western

blot and GST-fusion proteins by Coomassie Brilliant Blue (*CBB*) staining. The mean of quantifications corresponding to two independent experiments is shown (D, PLK1 intensity/GFP-BICD2 intensity in the immunoprecipitates; E, GFP-BICD2 intensity/GST-PBD intensity in the pulldowns). **F**, **G** In vitro phosphorylation of GST-BICD2 fragments by PLK1. Purified bacterial GST-BICD2 [1–271] (left) or GST-BICD2 [272–540] (right), were incubated with recombinant PLK1 plus [γ-$^{32}$P]ATP/Mg$^{2+}$ at 25 °C for the indicated times. ß-casein was used as a positive control. Phosphorylation was visualized and quantified as above. Relative $^{32}$P incorporation into BICD2 is shown in (**G**). Note that phosphorylation of GST-BICD2 [1-271] by PLK1 was extremely fast, occurring during the mixing of the reaction components and before it was stopped with electrophoresis sample buffer (*t* = 0). Non-radioactive replicates of the reactions at 15 min were used to identify phosphorylated sites by LC–MS/MS. Analysis using trypsin covered ~90% of the sequence in both cases. **H** Sites phosphorylated in vitro in BICD2 [1–271] and BICD2 [272–540] by PLK1, as identified by LC–MS/MS (see also Supplementary Fig. S2). *Conserved in H.s.*, conservation in humans (the orthologous residue is indicated between parenthesis). In vivo, sites previously determined to be phosphorylated in vivo in human cells (see https://www.phosphosite.org/). *PLK1 motif*, presence of a PLK1 consensus motif around the phosphorylation site ([D/E]X[S/T]Φ, where X is any aminoacid and Φ is an hydrophobic residue). Asterisks denote that a single phosphorylation site was detected that could not be assigned unequivocally among the two residues. **I** *Top*, schematic representation of BICD2, noting regions interacting with different proteins or protein complexes, plus the different fragments and residues mentioned in this figure (*Dynactin p.e., dynactin pointed end*). Residue number corresponds to mouse BICD2. Coiled-coil regions have been predicted using paircoil2[76]. *Bottom*, sequence alignments of residues surrounding Ser102, Thr319, Ser320, Thr321 and Ser331 in different BICD2 vertebrate orthologues are shown: mouse (Q921C5), human (Q8TD16), *Xenopus laevis* (Q5FWL8), *Dario rerio* (X1WDT9) and *Carcharodon carcharias*, a cartilaginous fish (XP_041047193). See also Supplementary Fig. S2 for similar alignments with BICD family members.

used the ability of recombinant BICD2 to affect the localization of different organelles when artificially tethered to them. We fused different BICD2 forms to the mitochondria-targeting domain (MTD) of a splice variant of *Drosophila* centrosomin[40]. When expressed in cells, the MTD fusion proteins co-localized with mitochondria, allowing us to assess dynein activity by observing mitochondria clustering around the centrosome, where microtubule minus-ends converge. As expected from previous observations[9], expression of GFP-BICD2 [1–575]-MTD resulted in a dramatic clustering of the mitochondria in most cells (85% of cells with clustered mitochondria, Fig. 3E). Clustering of mitochondria was around centrosomes as expected (see Supplementary Fig. S3A). Wild-type BICD2 (GFP-BICD2 WT-MTD) had a smaller effect, although mitochondria clustering was still detected in about half of the cells (50%), suggesting that in the conditions used a fraction of wild-type GFP-BICD2-MTD was able to form active dynein complexes. GFP-BICD2 S102A-MTD showed a slightly diminished ability to cluster mitochondria (43%) as compared to the wild-type form. Importantly, GFP-BICD2 S102D-MTD was significantly more efficient than wild type or S102A BICD2 in inducing the clustering of mitochondria (61% of cells with clustered mitochondria), thus supporting the hypothesis that the phosphorylation of Ser102 favors the formation of an active dynein motor complex.

### Modification of Ser102 induces structural changes in BICD2 that promote the formation of an active motor complex

We reasoned that the modification of Ser102 could regulate the interaction of BICD2 with dynein by increasing its affinity for the motor complex, directly affecting BICD2 overall conformation or both. To test the first possibility, we mutated Ser102 in BICD2 [1–575], devoid of regulation by the C-terminal part of the molecule. We observed that GFP-BICD2 [1–575; S102D] interacted with similar amounts of dynein as GFP-BICD2 [1–575] WT (or S102A) (Supplementary Fig. S3B). Moreover, when expressed as GFP-MTD fusion proteins, both wild type and S102D forms of BICD2 [1–575] induced the clustering of mitochondria to an almost identical extent (Supplementary Fig. S3C). This strongly

suggests that the modification of Ser102 has an effect only in the context of the full-length molecule and does not directly increase the intrinsic affinity of the N-terminal part of BICD2 for dynein.

We therefore sought to determine whether the modification of Ser102 could influence the structural architecture of BICD2. For this we expressed TwinStrep-BICD2 WT and TwinStrep-BICD2 S102D in insect cells and purified both recombinant proteins to homogeneity. The two forms of BICD2 showed almost identical profiles by size-exclusion chromatography combined with multiple angle light scattering (SEC-MALS), with a predicted MW of ~190 kDa, thus showing that the mutation did not have an effect on dimerization and gross molecular size (Supplementary Fig. S4A). This was confirmed by dynamic light scattering (DLS), that indicated that both proteins had a similar hydrodynamic radius of ~8 nm (Supplementary Fig. S4B). Also, we did not detect significant differences between wild type and mutated BICD2 when we measured their intrinsic fluorescence during a temperature ramp, suggesting that the mutation does not cause large structural rearrangements (Supplementary Fig. S4C).

To assess the structural disposition of both forms we used structural predictions by AlphaFold2[41] and AlphaFold-multimer[42] (thereafter AF) as well as experimental images of BICD2 obtained using electron microscopy. AF indicated that BICD2 likely comprises several coiled-coils (predicted with high confidence) connected by additional regions and loops (that were predicted with low confidence, see Supplementary Fig. S5A, B). This agreed with what had been previously proposed for BICD-family members[14] and suggested that it might be complicated to simulate how the coiled-coils organize in 3-dimensions. Still, predictions suggested that dimerization takes place head-to-head (Supplementary Fig. S5C, D), which agrees with cryoEM and crystal structural of some fragments of BICD2[11,43], and that the C-terminal domain forms a convoluted structure. It is noteworthy that predictions suggest that the C-terminal CC3 dimers can form a head to tail interaction with N-terminal CC1 dimers, implying that the coiled-coils are not placed in their maximum length but fold over the protein, and furthermore that residue S102 is actually located in a relatively

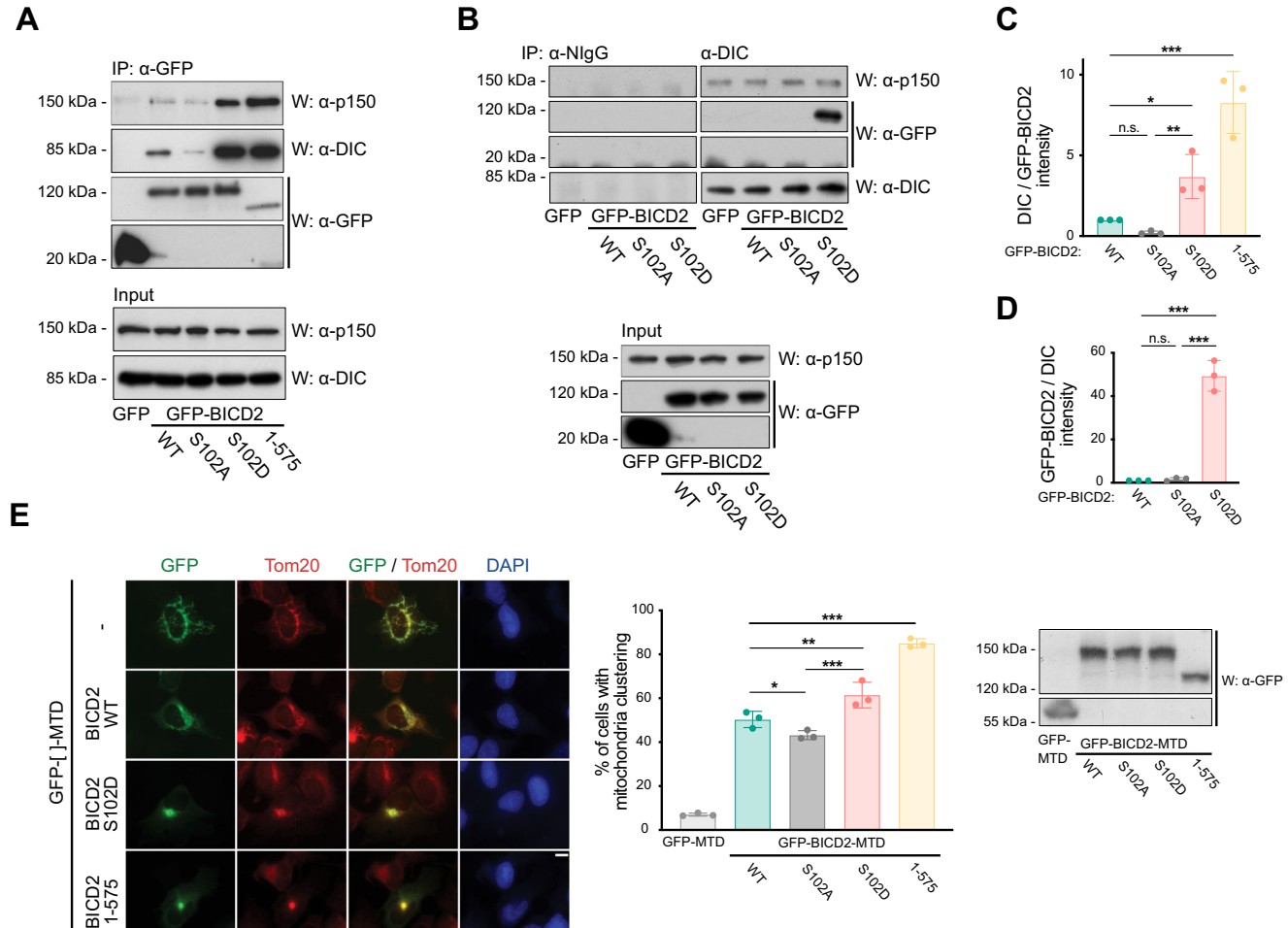

**Fig. 3 | Modification of BICD2 at Ser102 is able to control dynein binding and mobility. A–D** A phosphomimetic mutation in BICD2 Ser102 (BICD2 S102D) results in increased dynein and dynactin binding to BICD2. **A** The indicated GFP-fusion proteins were immunoprecipitated (*IP*) from HeLa cells and analyzed by western blot (*W*) to detect dynein and dynactin (using respectively anti-dynein intermediate chain, DIC, and anti-p150 antibodies), plus GFP. DIC and p150 levels in the corresponding extracts are shown in the lower panel. **B** Normal IgG (NIgG) and anti-DIC immunoprecipitates from extracts expressing the indicated GFP-fusion proteins proved by western blot (W) with the indicated antibodies. Levels of GFP-fusion proteins plus p150 in the corresponding extracts are shown in the lower panel. **C, D** Quantification of the previous results. Top, DIC intensity/GFP-BICD2 intensity in the immunoprecipitates; mean ± SD of three independent experiments (corresponding to **A**). Bottom, GFP-BICD2 intensity/DIC intensity in the immunoprecipitates; mean ± SD of two independent experiments (corresponding to **B**). Statistical significance analyzed using one-way ANOVA with post hoc analysis (no correction for multiple comparisons, Fisher's LSD test). In (**C**), WT vs. S102A, $P = 0.4380$ (n.s.); WT vs. S102D, $P = 0.0232$ (*); WT vs. 1-575, $P < 0.0001$ (***); S102A vs. S102D, $P = 0.0068$ (**). In (**D**), WT vs. S102A, $P = 0.8631$ (n.s); WT vs. S102D,

$P < 0.0001$ (***); S102A vs. S102D, $P < 0.0001$ (***). **E** A phosphomimetic mutation in BICD2 Ser102 (BICD2 S102D) results in increased dynein mobility towards the centrosomes as detected by mitochondria relocalization and clustering. Different GFP-fusion proteins were targeted to mitochondria through a C-terminal mito-chondria-targeting sequence domain (MTD) in U2OS cells. Representative immu-nofluorescence images are shown for each polypeptide, stained for GFP, Tom20, as a mitochondrial marker, and DAPI. Note that the S102D mutation favors clustering in a similar manner to the constitutively active BICD2 N-terminus (BICD2 1–575). Scale bar, 10 μm. See also Supplementary Fig. S3A, showing that clustering happens around the centrosomes. Quantification of the percentage of cells with clustered mitochondria (as defined as cells in which most mitochondria are spatially grouped in one or few clusters in the cytoplasm) for each different GFP-fusion form is shown (*n* = 3 biological replicates, 50 cells each; individual replicate means plus mean of replicates ± SD are shown; statistical significance analyzed using a Chi square test, with a two-sided *P* value; WT vs. S102A, $P = 0.0464$ (*); WT vs. S102D, $P = 0.0010$ (**); WT vs. 1–575, $P < 0.0001$ (***); S012A vs. S102D $P < 0.0001$ (***)). Expression levels of the different polypeptides as detected by western blot are shown.

proximal position to the place of this intramolecular interaction and to the C-terminal end of BICD2.

These predictions were consistent with the overall shape of BICD2 images in negative stain electron microscopy (EM) (Fig. 4A, B). For this EM analysis, we extracted several thousand individual molecule images of wild-type BICD2 and the S102D mutant from the micrographs, and these were subjected to unsupervised and reference-free image clas-sification and 2D averaging. In the electron microscope, wild-type BICD2 appeared mostly as comprising two arms connecting at one end (68% of the total selected particles). These triangularly-shaped images are similar to the compact conformation described recently for BICD2 using EM[44], where coiled-coils are not placed one after the other and

thus occupying their maximum possible length (Fig. 4B). Other less abundant images with a rod-like shape were also found (32% of the total selected particles). These are similar to the partially extended conformation described for BICD2[44].

Strikingly, when the S102D mutant was analyzed using the same conditions and methodologies, molecules in the compact conforma-tion were found to be less abundant (25% of the total selected parti-cles) whereas rod-shaped molecules were now the major species (75% of the total selected particles). This could be interpreted as indicating that the mutation facilitates the transition of BICD2 from a compact to a partially extended conformation, mimicking the conformational changes observed by EM at different pH[44]. Images for the compact

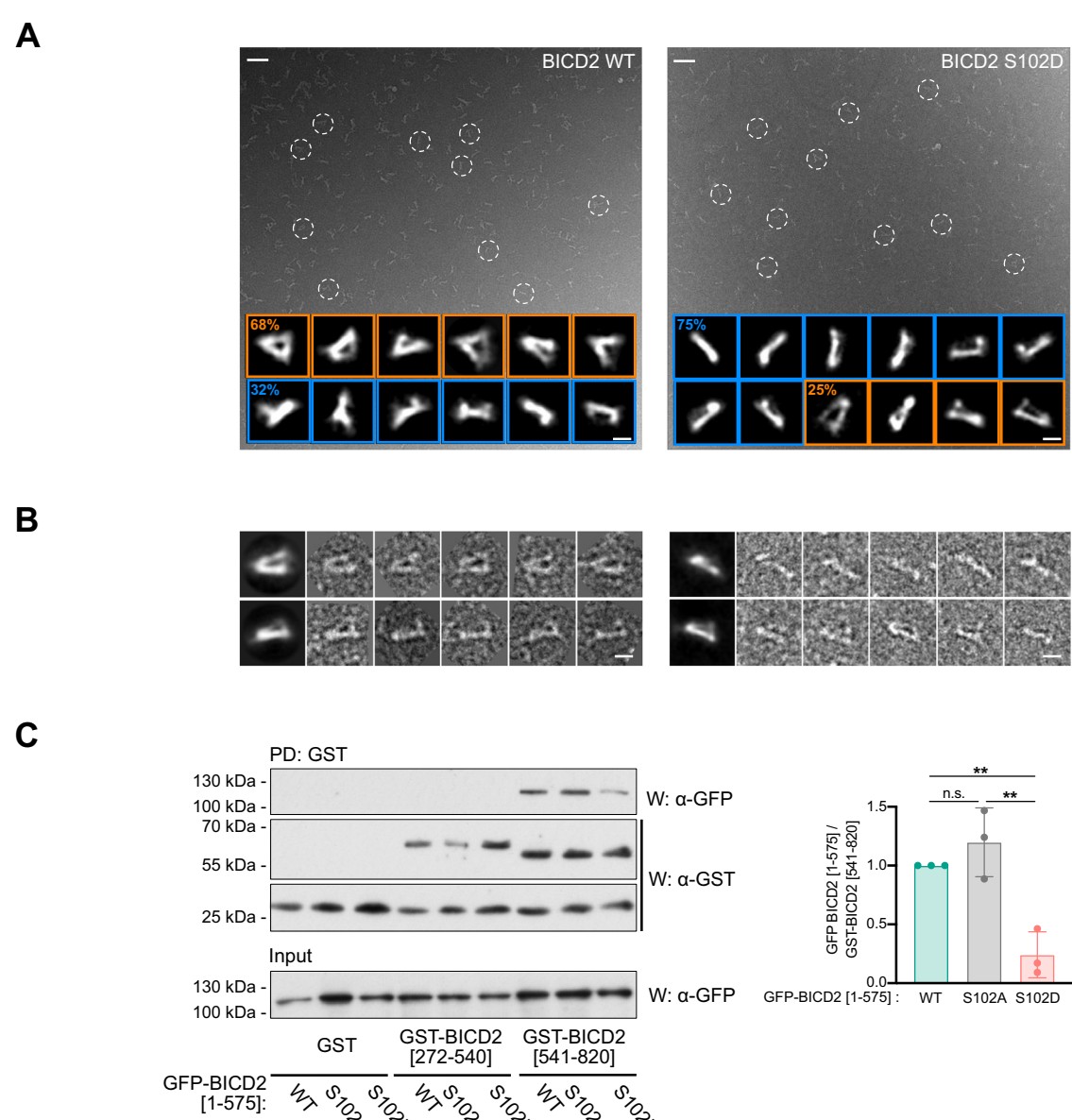

**Fig. 4 | Modification of Ser102 alters the conformation of BICD2.**
**A** Representative electron micrographs obtained for wild type (*left*) and S102D (*right*) TwinStrep-BICD2, using negatively stained samples. Several BICD2 individual molecules are highlighted within white dashed circles. Representative 2D average images are displayed at the bottom of each micrograph. The percentage of particles with a triangular or a rod-like morphology is noted in each case (30,476 BICD2 wild-type particles and 22,021 BICD2 S102D particles). Scale bars for micrographs and individual/average images represent 50 and 10 nm respectively. **B** Gallery of representative individual images of the wild type (left) and S102D (right) TwinStrep-BICD2 plus the corresponding average images. Scale bars, 10 nm. **A** and **B** Show results from one out of three experiments with identical results, in which 195 micrographies were acquired for BICD2 WT and 165 for BICD2 S102D. A representative image field is shown for each BICD2 form in (**A**). **C** Modification of

Ser102 interferes with the interaction between BICD2 N- and C-terminal regions. HeLa cells were transfected with different forms of BICD2 N-terminus (BICD2 [1–575]) plus a fusion of GST and the C-terminal part of BICD2 (BICD2 [540–820]). GST alone or fused to an internal BICD2 region (GST-BICD2 [272–540]) were used as controls. GST pulldowns (*PD*) were analyzed by western blot (*W*) using either anti-GFP or anti-GST antibodies. Note that expression of GST-tagged forms of BICD2 fragments resulting also in the apparition of free GST in the samples, possibly as a result of degradation. GFP polypeptides in the corresponding extracts are shown in the lower panel. Mean ± SD of quantifications corresponding to three independent experiments is shown. Statistical significance analyzed using one-way ANOVA with post hoc analysis (no correction for multiple comparations, Fisher's LSD test; WT vs S102A, $P = 0.2777$ (n.s.); WT vs. S102D, $P = 0.0038$ (**); S102A vs. S102D, $P = 0.0012$ (**)).

conformation are still present in the S102D mutant, suggesting that the mutation changes the equilibrium between conformations but several conformations coexist. The S102D mutation does not induce a fully extended conformation, which would reflect in much longer molecules in the microscope, and this agrees with our results using several biophysical techniques that indicate that the S102D mutation does not cause the large conformational changes expected for the fully extended conformation (see above).

Although our favored interpretation is that the S102D mutation helps the transition from a compact to a partially extended conformation, we cannot fully rule out that the triangularly-shaped and rod-shaped images could just represent a different view of BICD2 imaged in the microscope from a different angle, and that the S102D mutation affects the ratio of each view of the molecule detected in EM after the interaction with the carbon-coated support film. Subtle conformational changes in S102D could be sufficient to alter how the

protein binds to the EM support. Even in this scenario, the differences observed in EM between the wild type and the mutant can only be a consequence of modifications, even if subtle, in the structure of the protein caused by the phosphomimetic mutation.

After our EM observations, we sought to determine whether the modification of Ser102 could change the conformation of BICD2 by interfering with the interaction between its N-terminal and C-terminal parts. For this a GST-tagged fragment of BICD2 that comprises the CC3 region (GST-BICD2 [541–820]) was coexpressed with different GFP-tagged forms of BICD2 [1–575], containing both the CC1 and CC2 regions (see Fig. 2I). We observed that GST-BICD2 [541–820] (but not GST or GST-BICD2 [272–540], containing the CC2 region) was able to specifically pull down GFP-BICD2 [1–575] from cell extracts (Fig. 4C). We interpret this as replicating the intramolecular interaction between the N- and C-terminal regions of the adaptor in the context of the full-length molecule, and as was expected from previous two-hybrid data[14]. Strikingly, the phosphomimetic form of the N-terminus (GFP-BICD2 [1–575; S102D]), but not the S102A form, showed an impaired ability to interact with the C-terminus of BICD2. This effect was enhanced when high salt concentrations were used during washes of the pulldowns (Supplementary Fig. S6).

Altogether, we conclude that the phosphorylation of Ser102 directly interferes with the intramolecular interaction between the N-terminus and the C-terminus within the dimeric BICD2 molecule. This may not lead to a major alteration of the architecture of the BICD2 dimer, but may facilitate a conformation that is compatible with its binding to dynein and dynactin, resulting in the formation of an active motor complex.

### Dynein binding to BICD2 in G2 and M depends on PLK1 activity

Our results suggested that by phosphorylating BICD2 at Ser102, PLK1 may be able to regulate the formation of functional complexes between BICD2 and dynein in the G2 and M phases of the cell cycle, when the kinase is active. To explore this we determined the amount of dynein interacting with BICD2 in extracts from exponentially growing and G1/S-arrested cells, compared to extracts from cells progressing unimpaired through G2, cells arrested in G2 after CDK1 inhibition with RO-3306[45], and G2 cells in which both CDK1 and PLK1 where inhibited through the use of RO-3306 plus the PLK1 inhibitor BI 2536[46]. Cell cycle phase assignments were confirmed by flow cytometry analysis of DNA content (Supplementary Fig. S7A). Figure 5A shows that dynein readily coimmunoprecipitated with BICD2 in G2, with amounts that were slightly increased compared to the amounts precipitated in extracts from exponentially growing cells, and much higher than those observed in G1/S-arrested cells. G2 cells devoid of CDK1 activity showed diminished but still sizable interaction. Importantly, acute treatment (2 h) with BI 2536 of G2-arrested cells totally abrogated the interaction between BICD2 and dynein. Our results thus show that the interaction between BICD2 and dynein in G2 is strictly dependent on PLK1 activity but only partially dependent on the activity of CDK1, suggesting that another kinase such as CDK2 may be able to prime BICD2 for PLK1 binding. Indeed, treatment of RO-3306-arrested cells with the pan-CDK inhibitor roscovitine[47] completely abrogated the interaction, supporting this hypothesis.

To further assess the importance of PLK1, we chronically inhibited the kinase with BI 2536, a treatment that results in prometaphase arrest (Fig. 5B). BICD2 immunoprecipitates from cells treated with the PLK1 inhibitor were not associated with any observable amount of dynein. In contrast, treatment of the cells with STLC, an inhibitor of the kinesin-5 Eg5/KIF11 that equally arrests cells in prometaphase[48] (see Supplementary Fig. S7A), did not disrupt the interaction between dynein and BICD2. Of note, in early mitosis BICD2 shows a high apparent molecular weight that is not observed when the activity of PLK1 is inhibited. (Fig. 5B, arrowheads), suggesting that the kinase is responsible for a significant part of the phosphorylation of BICD2 in that phase of the

cell cycle. Acute treatment with BI 2536, RO-3306 or roscovitine did not interfere significantly with the amount of dynein bound to BICD2 in exponentially growing cells (Supplementary Fig. S7B). Altogether we interpret our results as indicating that PLK1, together with CDK1 and other CDKs, regulates the formation of BICD2/dynein complexes exclusively in G2 and M, and that this is regulated through the action of other mechanisms (e.g. binding to Rab6) in other phases of the cell cycle.

### BICD2 and dynein recruitment to the nuclear envelope is dependent on PLK1 activity and BICD2 Ser102 modification

We finally sought to reveal the functional significance of BICD2 phosphorylation in the context of the physiological roles of the adaptor in G2 and M. For this we first studied how the activity of PLK1 affected the recruitment of BICD2 and dynein to the nuclear envelope. As expected from previously published work[20,21], essentially all control HeLa cells in G2 showed BICD2 and dynein at the nuclear envelope (Fig. 6A, B). Most of them had high perinuclear levels of both proteins. Incubation with roscovitine strongly reduced both BICD2 and dynein staining at the nuclear envelope as described before[21]. RO-3306, which selectively inhibits CDK1 (>10-fold selectivity vs. CDK2[45]) only partially displaced BICD2 and dynein from the nuclear envelope, consistent with the notion that another CDK active in G2 such as CDK2 could have an important role alongside CDK1 in the regulation of their recruitment. Importantly, PLK1 inhibition with BI 2536 had a strong effect on both BICD2 and dynein localization in G2, resulting in more than 50% of cyclin B-positive cells with no apparent BICD2 or dynein localization at the nuclear envelope. In addition, most cells that retained some BICD2 or dynein perinuclear staining almost entirely had low levels of both the adaptor and the motor at the nuclear envelope.

We next sought to determine whether these observations resulted from PLK1 phosphorylation of BIDC2 at Ser102. For this we studied the ability of different recombinant forms of BICD2 to rescue the RNAi-mediated downregulation of endogenous BICD2 in G2 HeLa cells (Fig. 6C–F). As described[20], depletion of BICD2 strongly reduced the localization of dynein at the nuclear envelope in cyclin B-positive cells (Fig. 6C, F). This effect was reverted upon expression of GFP-BICD2 wild type, which was able to localize to the nuclear envelope (unlike GFP alone, Fig. 6C, E; see 6D for protein expression levels). GFP-BICD2 AAA (not phosphorylatable by CDK1 and thus unable to bind PLK1) and GFP-BICD2 S102A did not rescue endogenous BICD2 depletion, resulting in cells that either lacked dynein or had low levels of the motor at the nuclear envelope (Fig. 6C, F). Both phosphonull mutants showed a disrupted localization to the nuclear envelope in G2 (Fig. 6C, E), suggesting that besides regulating binding to dynein, PLK1 phosphorylation of BICD2 controls the interaction of the adaptor with the nuclear envelope (see below). Importantly, the phosphomimetic GFP-BICD2 S102D behaved similarly to the wild-type form, both localizing to the nuclear envelope and supporting the recruitment of dynein to this site (Fig. 6C, E, F). Of note, the phosphomimetic mutation of Ser102 was able to rescue dynein localization to the nuclear envelope even in a BICD2 form that cannot be phosphorylated by CDK1 (BICD2 [S102D, T319A, S320A, T321A]; Supplementary Fig. S8A). This mutation can also rescue dynein binding, suggesting that the exclusive role of $_{319}TST_{320}$ phosphorylation is to allow recruitment of PLK1 and the subsequent modification of Ser102 (Supplementary Fig. S8B).

### Centrosome tethering to the nucleus in G2 and separation in early mitosis depend on BICD2 Ser102 modification

Altogether our results show that the phosphorylation of BICD2 Ser102 by PLK1 is a key step for controlling dynein recruitment to the nuclear envelope in G2. The motor has been described to tether the centrosomes to the nucleus in G2 and early M[20,23–27], when it also contributes to the forces that will move the centrosomes apart[27,49]. We have shown in the past that PLK1 is necessary for normal centrosome separation

**A**

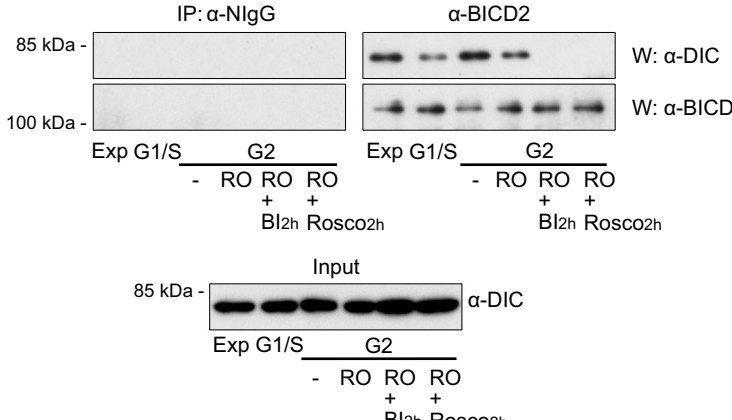
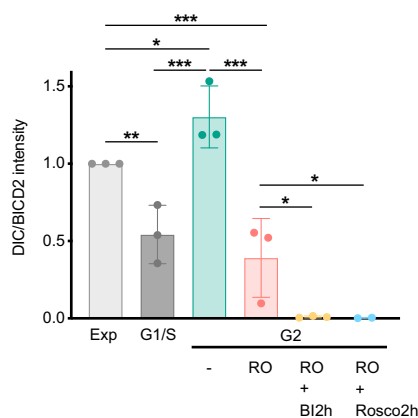

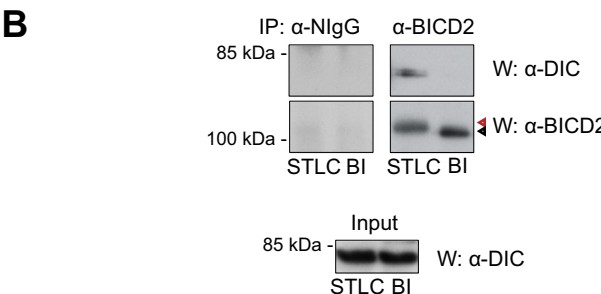

**B**

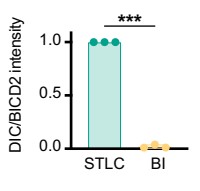

**Fig. 5 | BICD2 forms a complex with dynein in G2 and M that is dependent on PLK1 activity. A** PLK1 inhibition disrupts the interaction between BICD2 and dynein in G2. Normal IgG (*NIgG*) or anti-BICD2 immunoprecipitates (*IP*) from HeLa cells at different phases of the cell cycle were analyzed by western blot (*W*) using anti-DIC or anti-BICD2 antibodies. DIC levels in the corresponding extracts are shown in the lower panel. *Exp* exponentially growing cells, *G1/S* cells arrested at the G1/S border after a double thymidine block, *G2* cells in G2 (-, untreated G2 cells, 8 h after being released form a double thymidine block, *RO* RO-3306-arrested G2 cells (9 M, 16 h); *RO + BI2h* RO-3306-arrested G2 cells treated for 2h with 100 nM BI 2536, *RO + Rosco2h* RO-3306-arrested G2 cells treated for 2h with 55 μM roscovitine). Cell cycle assignation was confirmed by FACS (see Supplementary Fig. S7). The mean ± SD of quantifications corresponding to three independent experiments (except for *RO + Rosco2h*, that correspond to two independent experiments) is

shown (statistical significance analyzed using one-way ANOVA with post hoc analysis; no correction for multiple comparations, Fisher's LSD test; Exp vs G1/S, $P = 0.0049$ (**); Exp vs. G2/-, $P = 0.0408$ (*), Exp vs. G2/RO, $P = 0.0007$ (***); G1/S vs. G2/-, $P = 0.0001$ (***); G2/- vs. G2/RO, $P < 0.0001$ (***); G2/RO vs. G2/RO + BI2h, $P = 0.0138$ (*); G2/RO vs. G2/RO + Rosco2h, $P = 0.0222$ (*)). **B** PLK1 inhibition also disrupts the interaction between BICD2 and dynein in mitosis. As in (**A**). *STLC*, cells arrested in prometaphase with STLC (5 μM, 16 h); *BI*, cells arrested in prometaphase with BI 2536 (100 nM, 16 h). Note the increased apparent molecular weight of BICD2 in STLC- but not in BI 2536-arrested cells (red arrowhead, high MW; black arrowhead, low MW). The mean ± SD of quantifications corresponding to three independent experiments is shown (DIC intensity/BICD2 intensity in the immunoprecipitates; statistical significance analyzed using an unpaired *t*-test, with a two-sided *P* value; STLC vs. BI, $P < 0.001$ (***)).

and that this is at least partially the result of its role in indirectly controlling the phosphorylation of the kinesin Eg5 and its localization to the centrosomes[50]. We asked whether BICD2 Ser102 phosphorylation may also have a role during centrosome separation. We have observed the timing of centrosome separation to be variable among different cell types[51]. In G2, HeLa cells showed centrosomes that were adjacent or very close to the nucleus but still unseparated (Fig. 7A-C plus Supplementary Fig. S9). Downregulation of BICD2 resulted in bigger centrosome to nucleus distances, with few cases in which the centrosomes were as far as 15–20 μm away from the nucleus (mean distance of 4.1 μm vs. 1.6 μm in controls, Fig. 7B). This was not accompanied by centrosome separation since in all cases the two centrosomes remained in close proximity (Fig. 7C). Expression of GFP-BICD2 completely rescued the BICD2 depletion phenotype, with practically all cells showing centrosomes that were adjacent to the nucleus (mean distance of 0.2 μm). Expression of GFP-BICD2 [T319A, S320A, T321A] or GFP-BICD2 S102A failed to rescue centrosome to nucleus distances as effectively as the wild-type counterpart (mean distances of 2.0 μm and 2.1 μm, respectively). In contrast, expression of GFP-BICD2 S102D

resulted in distances that were similar to those observed for wild-type BICD2 (mean distance of 0.7 μm). These results confirm that, as expected from our observations in Fig. 6, the modification of BICD2 Ser102 is involved in controlling centrosome tethering to the nucleus in G2.

We next inquired whether the failure to modify Ser102 would also result in abnormal distance between centrosomes at the beginning of mitosis, when centrosomes separate in HeLa cells (Fig. 7D–F plus Supplementary Fig. S9). In prophase, almost all control cells had separated centrosomes that were adjacent to the nucleus (8.8 μm mean distance between centrosomes; we considered centrosomes to be separated when the intercentrosomal distance was more than 2 μm). Downregulation of BICD2 detached centrosomes from the nucleus as in G2 (Fig. 7D). The organelles were separated (mean intercentrosomal distance of 9.8 μm, Fig. 7F) but with a distribution in the cytoplasm that appeared to be random. Wild-type GFP-BICD2 was able to completely rescue centrosome attachment to the nucleus, as well as separation (mean distance of 10.0 μm). Both GFP-BICD2 AAA or GFP-BICD2 S102A were only able to partially rescue centrosome

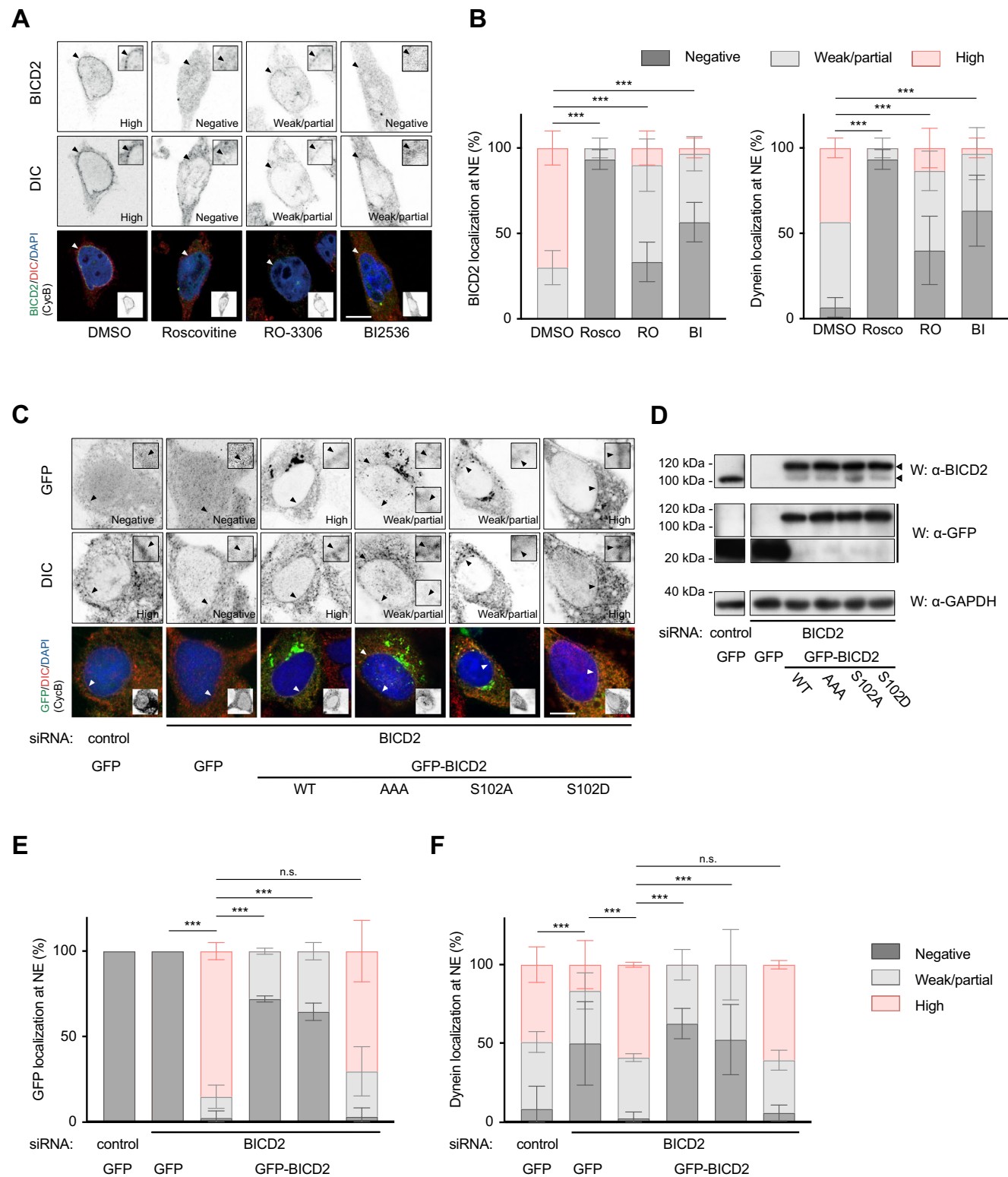

attachment to the nucleus and, strikingly, almost completely failed to support centrosome separation, respectively showing mean inter-centrosomal distances of 2.6 μm and 1.7 μm. In contrast, GFP-BICD2 S102D, was observed to completely rescue centrosome attachment to the nucleus (Fig. 7E), allowing for close to normal centrosome separation in prophase (mean intercentrosomal distance of 6.3 μm, Fig. 7F).

## Modification of Ser102 controls BICD2 binding to phosphorylated RanBP2 BBD

As mentioned above, our results strongly suggest that the phosphorylation of Ser102 (and the resulting change in conformation) not only favors the interaction of BICD2 with dynein and dynactin, but also controls the recruitment of the adaptor to the nuclear envelope (see Fig. 6C–E). We reasoned that Ser102 phosphorylation might directly

**Fig. 6 | PLK1 activity and BICD2 Ser102 phosphorylation are necessary for normal BICD2 and dynein localization at the nuclear envelope in G2. A, B** CDK and PLK1 inhibition strongly interfere with BICD2 and dynein localization to the nuclear envelope (*NE*) in G2 cells. **A** HeLa cells treated with DMSO, 55 μM Roscovitine (*Rosco*), 9 μM RO-3306 (*RO*) or 100 nM BI 2536 (*BI*) for 1 h were immunostained with the indicated antibodies plus DAPI. Example cells for each condition are shown. Scale bar, 10 μm. **B** G2 cells were scored for BICD2 and DIC at the nuclear envelope and the results quantified as shown (*n* = 3 biological replicates, 10 cells per experiment; mean ± SD is shown; statistical significance was analyzed using a Chi square test, with a two-sided *P* value; all comparisons *P* < 0.0001 (***)). **C–F** Phosphomimetic BICD2 S102D, but not BICD2 AAA or BICD2 S102A, is able to

rescue dynein nuclear envelope localization in G2 cells with low endogenous BICD2 levels. **C** HeLa cells were transfected with the indicated siRNAs and after 48 h transfected again with siRNAs plus the indicated GFP-tagged cDNA constructs. After 24 h cells were fixed and immunostained with the indicated antibodies plus DAPI. Example cells for each condition are shown (scale bar, 10 μm). **D** Expression levels of endogenous BICD2 and GFP-fusion proteins of a representative experiment. GFP-positive G2 cells were scored for GFP (**E**) and DIC (**F**) at the nuclear envelope and the results quantified as shown (*n* = 3 biological replicates, 8–15 cells per experiment; mean ± SD is shown; statistical significance was analyzed using a Chi square test, with a two-sided *P* value; all comparisons *P* < 0.0001 (***), except for WT vs S102D, *P* = 0.2724 (n.s.) in (**E**), and WT vs S102D, *P* = 0.7846 (n.s.) in (**D**)).

regulate the binding of BICD2 to the nucleoporin RanBP2, the "cargo" that recruits it to the nuclear surface. To test this and compare the behavior of BICD2 towards RanBP2 with that towards the canonical cargo Rab6, we expressed GST-fusion forms of both Rab6A and RanBP2 BICD2 Binding Domain (hereafter BBD, residues 2147-2287[20]), and observed their ability to bind purified forms of BICD2. Figure 8A shows that GST-Rab6 was able to interact with both wild type and S102D full-length BICD2. As expected, preloading the small G protein with a GTP analog (GTPγS) resulted in an increased binding (~5-fold, as compared to Rab6 preloaded with GDP). BICD2 S102 tended to bind slightly better than wild-type BICD2 to Rab6, either loaded with GDP or GTP, although our results in this regard were variable and the observed differences were consequently not significant. In contrast to Rab6, untreated GST-RanBP2 BBD did not significantly interact with neither wild type or S102D BICD2. CDK1 phosphorylation of the RanBP2 BBD has been shown to increase its ability to interact with the C-terminal region of BICD2[21], thus we preincubated GST-RanBP2 BBD with CDK1/cyclin B plus ATP, observing that the phosphorylation of RanBP2 BBD slightly improved the binding of full-length wild-type BICD2 to the BBD. Strikingly, BICD2 S102D strongly interacted with phosphorylated RanBP2 BBD (~12-fold increase as compared with wild-type BICD2). Our findings thus show that BICD2 phosphorylation by PLK1 not only controls its binding to dynein/dynactin, but also to its "cargo" RanBP2. And that RanBP2 has to be previously phosphorylated by CDKs to be able to interact with the phosphorylated adaptor (note that BICD2 S102D binds ~400 times better to phosphorylated RanBP2 BBD that BICD2 wild type to unmodified BBD).

Altogether our results highlight the importance of the phosphorylation of BICD2 by PLK1 in G2 and M and, moreover, identify adaptor phosphorylation as a previously unknown regulatory mechanism that controls recruitment and activation of the dynein machinery as well as cargo binding.

## Discussion

Dynein multiple functions suggest that the motor is subject to complex regulation in response to a variety of signaling inputs. Regulatory mechanisms can be direct, e.g. through post-translational modification of dynein different chains, or indirect, through the action of associated proteins. Indeed, cofactors such as dynactin, LIS1 and NDE1/NDEL1, and a number of structurally diverse cargo adaptors play a major role in modulating dynein activity[1,2]. Adaptors are particularly important in this regard. While connecting the dynein complex to specific cargos, as well as to other motors such as kinesins, they also effectively function as activators by facilitating the interaction between dynein and dynactin[1,7]. How this is regulated in response to different physiological demands is not well understood. One possibility is that the ability to activate dynein would be triggered after cargo binding to the adaptor, thus spatiotemporally linking motor activity to cargo availability. This regulatory mechanism has been proposed for BICD family members[8]. Both BICD1 and BICD2 are autoinhibited through an intramolecular interaction between their N- and C-terminal regions that is thought to physically impede the interaction with dynein and dynactin[10,14,52]. Binding of the small GTPase Rab6 to the C-terminus of BICD1/2 would

liberate the N-terminus, allowing it to organize an active motor complex in order to transport exocytotic vesicles[14,17,19]. Other adaptors such as BICDL1[53] or HOOK proteins[54] apparently interact with dynein and dynactin independently of cargo binding, thus suggesting that alternative means or regulation are at play.

In this work we put forward adaptor phosphorylation as an alternative mechanism to induce the assembly of active dynein complexes. We show that, in response to phosphorylation by the G2/M protein kinase PLK1, BICD2 is "activated", becoming able to bring together dynein and dynactin into an active motor complex (see Fig. 8B for a graphical depiction of this). In fact, dynein binding to BICD2 is almost completely dependent on the activity of PLK1 in G2 and M (but not other phases of the cell cycle). Phosphorylation is consequently the prevalent mechanism regulating the adaptor in these two phases. Accordingly, a change in BICD2 electrophoretic mobility suggest that most of the adaptor is phosphorylated at least in M.

We propose that BICD2 is subject to dual regulation. In this view, during G1 and S and in the context of lipid vesicle transport, the adaptor is controlled by the previously described Rab6-based mechanism. And in G2 and M, when a major reorganization of the intracellular membrane system occurs and BICD2 switches partner and interacts with RanBP2 at the nuclear pores, BICD2 is regulated by PLK1 phosphorylation. PLK1 activity is circumscribed to G2 and M, but our data suggests that the timing of this regulatory switch is doubly ensured by the additional need for a priming phosphorylation by a G2/M CDK, that facilitates PLK1 binding to BICD2. In vitro and possibly in conditions of excess PLK1 activity this priming step might not be required (e.g., we observed in vitro phosphorylation of BICD2 fragments by PLK1 independently of a priming phosphorylation). In cells, though, this seems to be a necessary step, since mutation of CDK sites ($_{319}$TST$_{321}$) is similarly deleterious to the G2 and M functions of BICD2 as is the mutation of the main PLK1 site (Ser102). Regarding the identity of the priming kinase in vivo, we observed that treatment of cells with RO-3306, a specific inhibitor of CDK1, does not completely abolish the formation of BICD2/dynein complexes. This suggests that other CDK family members active in G2 (i.e. CDK2) could have a role as PLK1 priming kinases. This is supported by the finding that the pan-CDK inhibitor roscovitine more completely eliminated BICD2/dynein complexes.

PLK1 is able to rapidly phosphorylate BICD2 at Ser102, a modification that we suggest has both structural and functional consequences. The residue is located at the predicted elongated N-terminal coiled-coil region, close to the dynein-interacting CC1 box (see Fig. 2I). Notably, a mutation of the nearby Ser107 found in a rare form of human spinal muscular atrophy (SMALED2; OMIM 615290) results in an enhanced ability to form motile dynein complexes, highlighting the importance of Ser102 and its surroundings for the regulation of BICD2[19]. Indeed, DeepMind-based methods for protein structure prediction[41] open the possibility that BICD2 Ser102 could be at or close to the interface in CC1 that interacts with the CC3 coiled-coil region and thus ideally suited to control BICD2 conformation. In fact our data show that Ser102 phosphorylation impairs the intramolecular interaction between the C- and the N-terminal regions of the dimeric

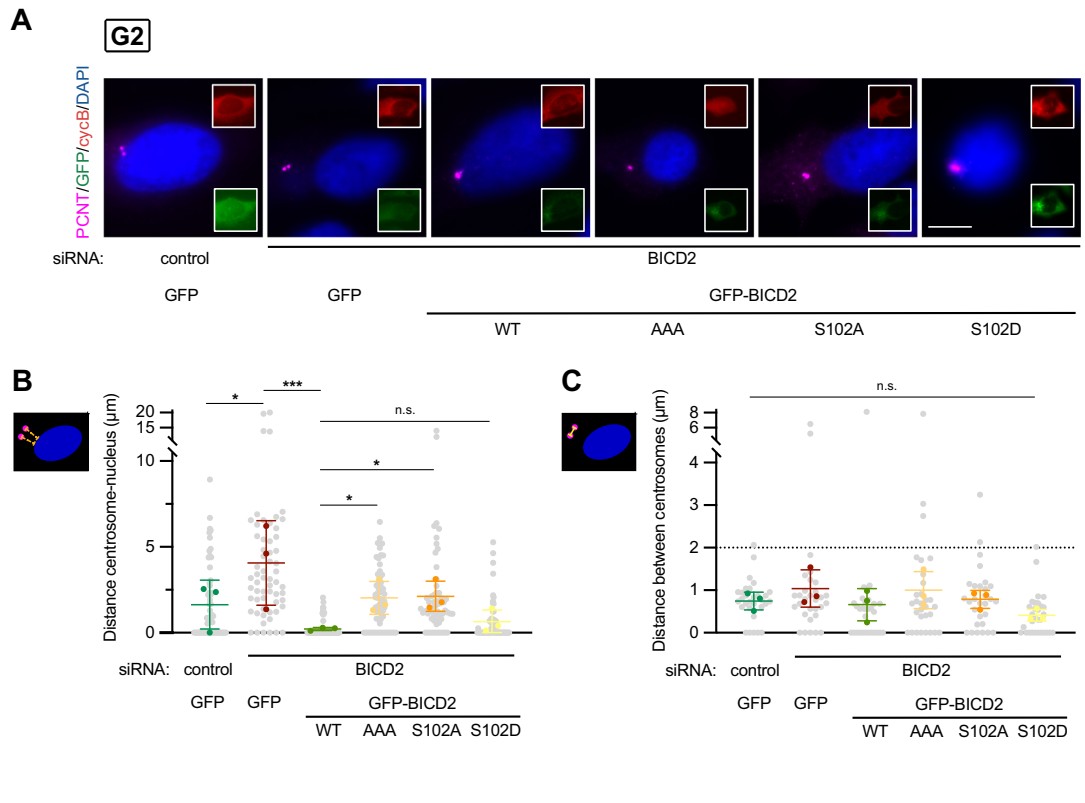

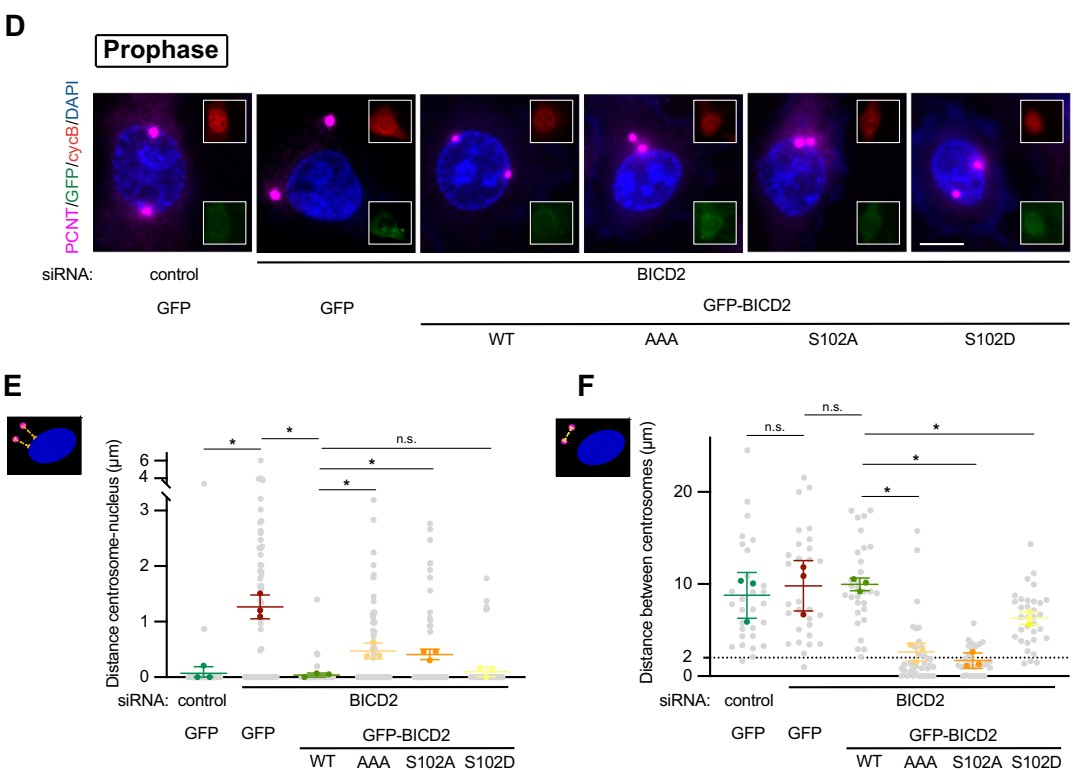

structure of the adaptor, proposed to be the basis of BICD2 autoinhibition[14]. This favors a structural change and a conformation that may be more amenable to interact with dynein/dynactin. Electron micrographs of wild-type BICD2 showed that the molecule is predominantly in a triangular conformation, similar to what has been described for *Drosophila* BICD[55,56], although in the mammalian homolog the looped part seems to comprise a bigger portion of the

molecule, similar to the recently reported compact conformation by Fagiewicz et al.[44]. This triangular shaped "compact conformation" might be a consequence of the inhibitory intramolecular interaction taking place between the adaptor N- and C-terminal parts. Phosphorylation of Ser102 introduces some structural alteration of the BICD2 molecule that favors either a more open conformation or some more subtle structural rearrangement that might result in exposition of one

**Fig. 7 | BICD2 Ser102 phosphorylation regulates centrosome tethering to the nucleus and separation during G2 and early mitosis.** The figure shows the ability of different BICD2 mutant forms to rescue centrosome tethering to the nucleus and centrosome separation upon endogenous BICD2 downregulation in G2 (**A**–**C**) and prophase (**D**–**F**). For this, HeLa cells were transfected with the indicated siRNAs and cDNAs as in Fig. 6C–F, fixed and immunostained with the indicated antibodies plus DAPI. Pericentrin was used as a centrosomal marker, cyclin B and DNA staining to identify cell cycle stage. Example cells for each condition are shown in (**A**) and (**D**). Insets show that cells are positive for cyclin B (red channel) and express GFP-tagged recombinant proteins (green channel). Note uncondensed DNA and mostly cyto-plasmatic cyclin B in G2 cells (in **A**) and condensed DNA and nuclear cyclin B in prophase cells (in **D**). Centrosomes are considered to be separated if the inter-centrosomal distance is more than 2 μm (indicated with a dashed line in the graphs in **C** and **F**). Scale bar, 10 μm. See also Supplementary Fig. S9 for the individual image channels. **A**–**C** Phosphomimetic BICD2 S102D, but not BICD2 AAA or BICD2 S102A, is able to rescue centrosome tethering to the nucleus in G2 cells with low endogenous BICD2 levels. Centrosome to nucleus (**B**) and centrosome to centro-some distances (**C**) were quantified in GFP-positive G2 cells ($n = 3$ biological repli-cates, 10 cells per experiment; individual replicate means plus mean of replicates ± SD are shown; statistical significance analyzed using one-way ANOVA with post hoc analysis; no correction for multiple comparations, Fisher's LSD test). In (**B**): control

RNAi/GFP vs BICD2 RNAi/GFP, $P = 0.0130$ (*); BICD2 RNAi/GFP vs. BICD2 RNAi/GFP-BICD2 WT, $P = 0.0007$ (***); BICD2 RNAi/GFP-BICD2 WT vs. BICD2 RNAi/GFP-BICD2 AAA, $P = 0.0483$ (*); BICD2 RNAi/GFP-BICD2 WT vs. BICD2 RNAi/GFP-BICD2 S102A, $P = 0.0395$ (*); BICD2 RNAi/GFP-BICD2 WT vs BICD2 RNAi/GFP-BICD2 S102D, $P = 0.6049$ (n.s.). In (**C**), $P = 0.2841$ (n.s.). **D**–**F** Phosphomimetic BICD2 S102D, but not BICD2 AAA or BICD2 S102A, is able to rescue centrosome separation in pro-phase cells with low endogenous BICD2 levels. Centrosome to nucleus (**D**) and centrosome to centrosome (**E**) distances were quantified in GFP-positive prophase cells ($n = 3$ biological replicates, 10 cells per experiment; individual replicate means plus mean of replicates ± SD are shown; statistical significance analyzed using one-way ANOVA with post hoc analysis; no correction for multiple comparations, Fisher's LSD test). In (**E**): control RNAi/GFP vs BICD2 RNAi/GFP, $P = 0.0169$ (*); BICD2 RNAi/GFP vs. BICD2 RNAi/GFP-BICD2 WT, $P = 0.0137$ (*); BICD2 RNAi/GFP-BICD2 WT vs. BICD2 RNAi/GFP-BICD2 AAA, $P = 0.0322$ (*); BICD2 RNAi/GFP-BICD2 WT vs. BICD2 RNAi/GFP-BICD2 S102A, $P = 0.0342$ (*); BICD2 RNAi/GFP-BICD2 WT vs BICD2 RNAi/GFP-BICD2 S102D, $P = 0.4665$ (n.s.). In (**F**): control RNAi/GFP vs BICD2 RNAi/GFP, $P = 0.7449$ (n.s.); BICD2 RNAi/GFP vs. BICD2 RNAi/GFP-BICD2 WT, $P = 0.9406$ (n.s.); BICD2 RNAi/GFP-BICD2 WT vs. BICD2 RNAi/GFP-BICD2 AAA, $P = 0.0124$ (*); BICD2 RNAi/GFP-BICD2 WT vs. BICD2 RNAi/GFP-BICD2 S102A, $P = 0.0110$ (*); BICD2 RNAi/GFP-BICD2 WT vs BICD2 RNAi/GFP-BICD2 S102D, $P = 0.0431$ (*).

or more of the dynein/dynactin binding sites. Future efforts aimed at determining the atomic structure of both wild type and S102D BICD2 should reveal the molecular basis of this conformational change, be it electrostatic or steric repulsion, or a registry shift[43]. But altogether our interpretation that the phosphorylation of Ser102 results in the functional opening of BICD2 is supported by biochemical data and cell-based assays.

The importance of BICD2 phosphorylation as a major regulatory mechanism for dynein is highlighted by its requirement for the recruitment of the motor to the nuclear envelope in the G2 and early M phases of the cell cycle. We show that in HeLa cells this depends on the action of both CDK1 (and possibly a second G2 CDK) and PLK1. CDK1 has been previously reported to control dynein recruitment though its ability to phosphorylate the nucleoporin RanBP2, thus favoring BICD2 binding to the nuclear pore. It also controls a second dynein-recruitment pathway connecting the motor to Nup133, through the phosphorylation of CENP-F and NDE1[21,57]. We reveal an additional independent layer of regulation that acts directly on the BICD2 adap-tor, possibly affecting dynein more broadly. Our observations not only are in agreement with the previously suggested involvement of PLK1 in regulating dynein nuclear envelope recruitment[21,58], but place the Polo-family kinase in a central position of this pathway.

Remarkably, our results with phosphonull mutants show that phosphorylation of BICD2 by CDKs and by PLK1 does not only controls the ability of the adaptor to bind dynein and recruit it to the nucleus, but also the nuclear recruitment of BICD2 itself. Using purified pro-teins we show that BICD2 needs to be phosphorylated and presumably in a more "open" conformation to be able to bind to phosphorylated[20] RanBP2 (and possibly other nuclear membrane proteins capable of interacting with the adaptor such as Nesprin-2[22]). In this view RanBP2 (or other nuclear membrane proteins) would not behave as a canonical cargo such as Rab6, which are thought to induce opening of BICD2 upon binding and allow it to interact with dynein. Instead, it would be the phosphorylation of BICD2 Ser102 that would change the con-formation of the adaptor, allowing it to bind to both dynein/dynactin *and* the nucleoporin (note that, to our knowledge, direct binding of unmodified BICD2 to RanBP2 has only been shown in vitro for the C-terminal region of the adaptor[20,21,59]). The fact that RanBP2 and Rab6 have different binding sites in BICD2[52,60], could possibly provide a molecular explanation for this hypothesis.

The need of a regulated interaction between BICD2 and RanBP2 in G2 may be understood in view of the reported affinity between the C-terminus of BICD2 and RanBP2, which is much higher than the affi-nity for Rab6(GTP)[61]. One can speculate that unregulated high-affinity

binding of RanBP2 to (full length) BICD2 would result in massive recruitment of the adaptor to the nuclear pore. In fact Noell et al. calculated that ~1000 BICD2 molecules per cell would be recruited to the nucleus by RanBP2 in the absence of regulation[61]. Therefore CDK1 and PLK1 phosphorylation and "opening" of BICD2 ensures that the adaptor interacts with the nucleoporin exclusively at the right cell cycle phase, something that will be further favored by CDK1 phos-phorylation of RanBP2[21]. A recent report has shown that an artificially dimerized form of a minimal BICD2 binding domain of RanBP2 (Nup358min-zip, containing residues 2148–2240) is capable to pro-duce active dynein/dynactin/BICD2 complexes[59]. RanBP2 phosphor-ylation could alter the dimeric conformation of the nucleoporin, favouring its binding to BICD2, but more work will be needed to clarify this point, as well as how it fits with ours and other published data.

Finally, we find that BICD2 phosphorylation controls centrosome tethering to the nucleus as well as centrosome separation in G2 and early mitosis (Fig. 8B, boxes). Regarding the importance of BICD2 for the positioning of centrosomes close to the nucleus, our results are similar to what has been reported by others[20,26,62] (the adaptor might be dispensable in prophase in some cell lines[27], this possibly being the result of variations in the strength of the later-acting Nup133/CENPF/NDE1/NDEL1 dynein recruiting pathway[26]). Importantly, our findings indicate that the phosphorylation of BICD2 is central to the regulation of the process.

We also implicate BICD2 and its phosphorylation in the control of centrosome separation. While dynein is definitely instrumental in the tethering of centrosomes to the nucleus in G2 and early M, its parti-cipation in the production of forces that results in the separation of the organelles during mitosis is less clear. In both *D. melanogaster* and *C. elegans* dynein is necessary for centrosome separation[23–25], but the motor has been reported to be dispensable for the separation of centrosomes in prophase in human U2OS and RPE-1 cells[20,27,49]. In these cells, the main role of dynein during separation would be to keep the centrosomes close to the nucleus, antagonizing (together with peri-nuclear actin[63]) the forces produced by Eg5, that push centrosomes far apart but also away from the nucleus. Puzzlingly, in cells with low levels of Eg5, (nuclear-associated) dynein has been shown to be involved in prophase centrosome separation[27], suggesting that a not completely understood relationship exist between dynein and Eg5 forces and the separation of the centrosomes during early mitosis.

We observed that the downregulation of BICD2 in prophase HeLa cells did not interfere with the apparent moving apart of the centro-somes. However, we note that without BICD2 centrosomes appeared to be randomly distributed in the cytoplasm, similar to what was

**A**

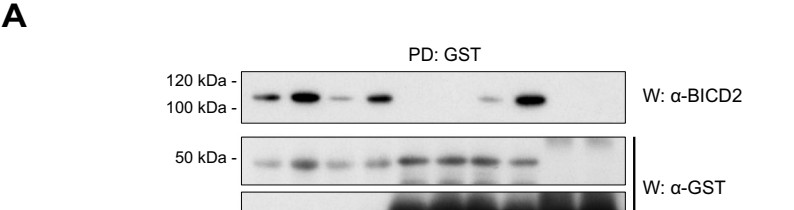

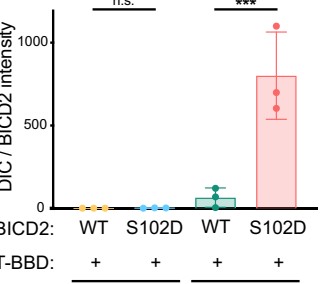

**B**

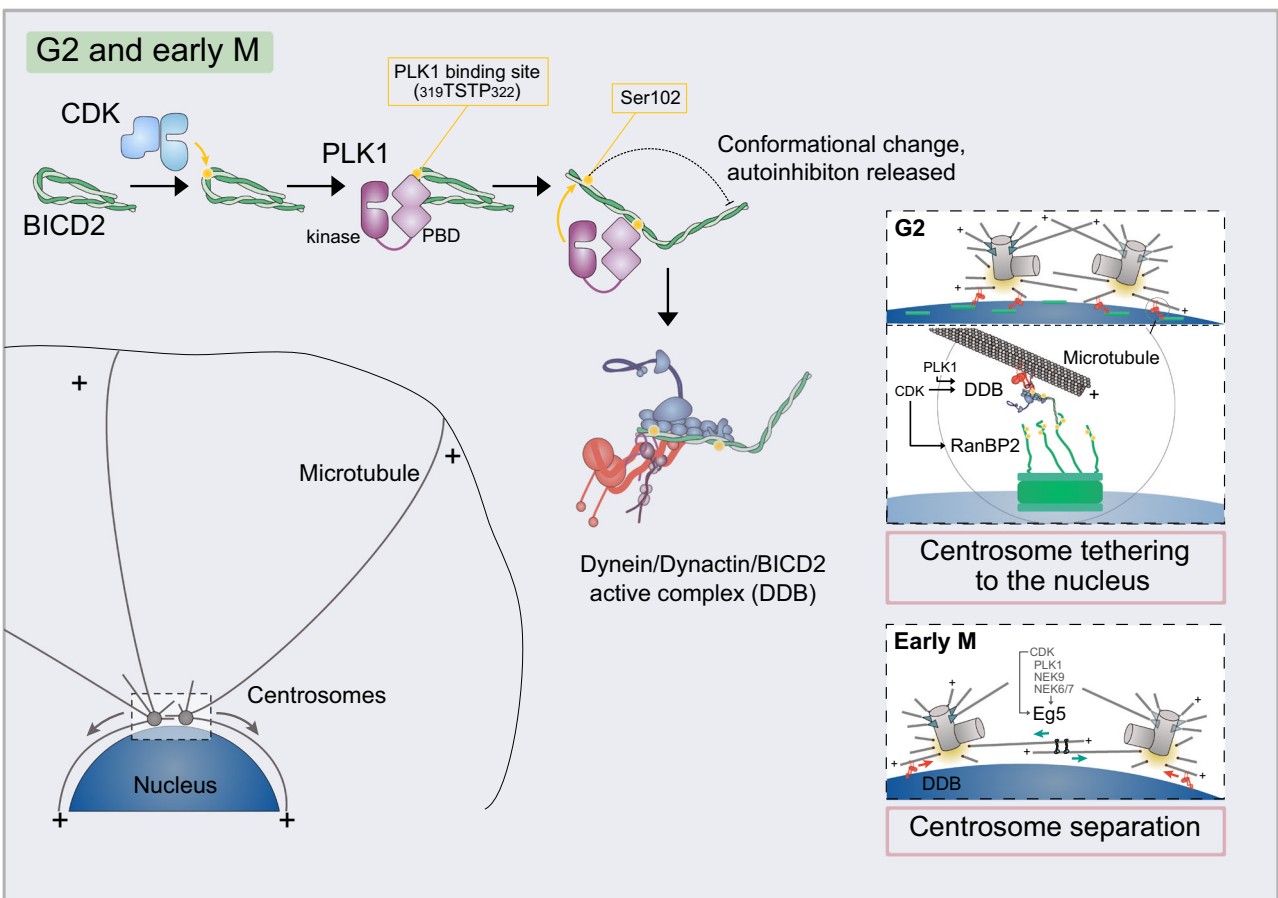

reported for treatments that disrupt the microtubule cytoskeleton[64,65]. Thus, in these conditions the centrosomes may not be actively separating but drifting apart as a result of losing contact with the nucleus. In any case, our results indicate that BICD2 and its phosphorylation have a previously unreported role during separation, as the expression of BICD2 phosphonull mutants interfered with the observed separation, resulting in small intercentrosomal distances,

while expression of BICD2 S102D rescued centrosome separation to an extent similar to that observed for the wild-type form of the adaptor. More experiments will be needed to clarify this.

We and others have previously shown that centrosome separation depends on the phosphorylation of Eg5 by CDK1 and NEK6/7, the latter kinases being activated by the related NEK9 downstream of PLK1 and CDK1[50,66]. Our results add a new node to this regulatory network, its

**Fig. 8 | Ser102 controls BICD2 binding to phosphorylated RanBP2 BBD.** Graphical model illustrating the role of phosphorylation in controlling BICD2 activity and function in G2 and M. **A** Wild type and S102D TwinStrep-BID2 were incubated with GSH agarose beads bound to GST-Rab6 (preloaded with either GDP or GTPγS) or GST-RanBP2 BBD (BICD2 Binding Domain, residues 2147-2287). When indicated, GST-RanBP2 BBD had been previously phosphorylated by incubation with CDK1/Cyclin B and ATP/Mg$^{2+}$. After washes, TwinStrep-BICD2 bound to the beads was detected by western blot (*W*) using anti-BICD2 antibodies. GST-fusion proteins were subsequently detected with anti-GST antibodies (note that GST-RanBP2 BBD expresses poorly and gets easily degraded, resulting in free GST in the samples). One of three experiments is shown, together with a quantification of the amount of BICD2 bound to RanPB2 BBD and Rab6 in the different conditions (mean ± SD of three independent experiments; BICD2 intensity/GST intensity in the pulldowns; statistical significance analyzed using one-way ANOVA with post hoc analysis; no correction for multiple comparations, Fisher's LSD test). For GST-BBD (top): WT vs. 102D, $P = 0.9912$ (n.s.); WT + CDK1 vs. 102D + CDK1, $P < 0.0001$ (***); for GST-Rab6

(bottom): WT + GTPγS vs. 102D + GTPγS, $P = 0.8952$ (n.s.); WT + GDP vs. 102D + GDP, $P = 0.8230$ (n.s.). **B** In G2 and early M, CDKs (CDK1 and possibly CDK2) phosphorylate BICD2 (yellow circles denote phosphorylation). This allows BICD2 to interact with the PBD domain of PLK1. In turn PLK1 phosphorylates BICD2 at Ser102, inducing a conformational change that results in the interaction of the adaptor with dynein and dynactin and the formation of an active DDB complex. In G2 (right, top box), phosphorylated BICD2 can interact with RanBP2 (also known as NUP358), once this nucleoporin has been phosphorylated by CDK1[21] (and possibly CDK2). RanBP2/BICD2[20], together with the Nup133/CENPF/NDE(L)1[26] dynein recruiting pathway, concentrates active dynein complexes to the nuclear pores (green boxes), effectively tethering the centrosomes to the nuclear envelope. Nesprin-2 additionally collaborates in recruiting BICD2/dynein to the nuclear membrane[22]. In early mitosis (lower box) this facilitates centrosome separation, mostly driven by the bipolar kinesin Eg5, that is also regulated by CDK1[66] and PLK1 through the action of the NIMA kinases NEK9, NEK6, and NEK7[50].

---

convoluted structure likely reflecting the importance of timely centrosome separation for the normal segregation of chromosomes later in mitosis, as centrosome separation before nuclear envelope breakdown has been shown to be important for proper chromosome attachment to the spindle[67].

Besides centrosome nuclear attachment and separation, other roles of perinuclear dynein in G2 and M, such as its involvement in nuclear envelope breakdown or in apical nuclear migration in neural progenitors, may also involve regulation through phosphorylation of BICD2. In neural progenitors, for instance, BICD2 has been shown to be key for the migration of the nucleus to the apical region in G2 before mitosis, a process mediated by dynein that is necessary for the normal division of these specialized cells[21,29,30]. We speculate that PLK1 may be regulating this at least partially through the phosphorylation of the adaptor.

Our work shows that the formation of active dynein complexes can be regulated not only through cargo binding to specific dynein adaptors, but also through adaptor phosphorylation. Further studies should establish whether other adaptors besides BICD2 share this regulatory mechanism. This may be the case for adaptors relying on autoinhibition for the control of their activity, such as the closely related BICD1 or *Drosophila* BICD. In fact BICD1 has a residue that is homologous to BICD2 Ser102, suggesting that it may be regulated by PLK1 (BICD1 is possibly regulated by other kinases such as GSK-3ß, that is able to modify its C-terminus in the context of the regulation of centrosomal components and aster microtubule dynamics in interphase[68]). *Drosophila* BICD is known to depend on phosphorylation for its roles during oocyte differentiation, although the details of this are unclear[69]. Interestingly, BICD lacks a phosphorylatable residue at the exact homologous position of BICD2 Ser102 but it shows a canonical putative PLK1 phosphorylation site 11 residues towards the C-terminus of the protein (not present in other BICD-family members), thus suggesting that it may be subject to Polo kinase regulation (see Supplementary Fig. S2, showing the relevant positions in different BICD family members).

Our results, summarized graphically in Fig. 8B, establish that phosphorylation has a central role in regulating dynein physiology. Indeed, not only cargo adaptors but most dynein subunits are phosphorylated in vivo (e.g. see[70], and https://www.phosphosite.org/), possibly by a number of protein kinases including PLK1[58,71]. The detailed study of this will surely expand our understanding of the functional complexity of the dynein motor complex.

## Methods

### Mammalian expression constructs and siRNAs

pEGFP-C2-BICD2, containing the cDNA for mouse BICD2, was a gift from Anna Akhmanova (Utrecht University) and is described in ref. 14. Fragments and mutant forms of BICD2 were produced

using QuickChange Lighting Site-Directed Mutagenesis kit (Agilent Technologies) according to the manufacturer's instructions using the following primers and the appropriate reverse complements: BICD2 [1–575]: 5′-GAGGGCCGCGGGTGACGGTCACCTGTC-3′; BICD2 [576–820]: 5′-GATCTCGAGAGAATTCGCCACCCGCCGGTCACCTGTCCTCTTG-3′;

BICD2 [T319A, S320A, T321A]: 5′-CATCCTTGGACAACAAGGCAGCCGCACCCAGGAAGGATGG-3′;

BICD2 S102A: 5′-GAGAGCCGGGAGGAGGCCCTGATCCAGGAGTCG-3′;

BICD2 S102D: 5′-GAGAGCCGGGAGGAGGACCTGATCCAGGAGTCG-3′.

To determine the mitochondrial clustering ability of the different BICD2 forms, the desired BICD2 cDNAs were amplified from pEGFP constructs by PCR using Phusion High-Fidelity DNA polymerase (Thermo Fisher) and inserted using a Gibson assembly strategy (NEB) into a pEGFP-C1-MTD plasmid, also containing the cDNA of the isoform G of centrosomin from *Drosophila melanogaster* (obtained from the Drosophila Genomics Resource Center, accession number AT09084). pEBG-BICD2 [272–540] and pEBG-BICD2 [541–820] were obtained by subcloning the corresponding cDNAs, amplified by PCR using Pfu Ultra HF DNA polymerase (Agilent Technologies) and pEGFP-C2-BICD2 as a template, into empty pEBG vector. Coding regions of all constructs was sequenced prior to use.

For endogenous BICD2 knockdown we used the following siRNA sequence, after ref. 20: 5′-GGAGCUGUCACACUACAUGUU-3′. A siRNA against luciferase with the following sequence was used as control: 5′-CGUACGCGGAAUACUUCGA-3′.

### Bacterial expression

pGEX-PLK1 PBD [345–603] was previously described[50]. pGEX- BICD2 [1–271] and pGEX-BICD2 [272–540] were obtained by subcloning the corresponding cDNAs, amplified by PCR using Pfu Ultra HF DNA polymerase and pEGFP-C2-BICD2 as a template, into empty pGEX-4T1. pGEX-BICD2 [272–540; T319A, S320A, T321A] mutant was produced using QuickChange Lighting Site-Directed Mutagenesis kit according to the manufacturer's instructions with the primers described above. Proteins were expressed in *Escherichia coli* BL21(DE3) pLysS cells and purified using glutathione agarose beads (GE Healthcare) according to the manufacturer's instructions. BICD2 fragments were eluted with 10 mM reduced L-glutathione (Merck Life Science). PLK1 PBD was kept bound to the agarose and stored at 4 °C for subsequent pull-down experiments.

pGEX-Rab6A, containing the cDNA of human Rab6A, was a gift from Anna Akhmanova (Utrecht University) and is described in ref. 17. RanBP2 BBD (residues 2147–2287) was cloned from a human kidney cDNA library (Clontech) and inserted into the PGEX 4T1 vector. Both proteins were expressed in *E. coli* Rosetta cells and purified using

glutathione (GSH) agarose beads as above. Proteins were kept bound to the agarose and stored at −20 °C in 50% glycerol for subsequent pull-down experiments.

## Baculovirus expression
pLIB-TwinStrep-3C-BICD2 was constructed using a Gibson assembly strategy with the corresponding BICD2 cDNA (amplified from pEGFP constructs by PCR using Pfu Ultra HF DNA polymerase), plus pOPINN1S3CTwinStrep-3C and pLIB plasmids. pLIB-TwinStrep-3C-BICD2 S102D was made using QuickChange Lighting Site-Directed Mutagenesis kit.

Full-length BICD2 proteins were produced by baculovirus-mediated expression in insect cells. Bacmids were generated by Tn7 transposition of pLIB-TwinStrep-3C-BICD2 constructs into the EMBacY baculovirus genome as described[72]. Baculoviruses were generated as described[73] with minor modifications.

Full-length BICD2 WT and BICD2 S102D were purified from infected cells using the N-terminal TwinStrep-tag. Frozen pellets were resuspended on ice in Lysis Buffer B (50 mM Tris pH 8.0, 300 mM NaCl, 2 mM β-mercaptoethanol, 0.1% IGEPAL CA-630, 2× complete protease inhibitor cocktail EDTA-free (Roche), 5% glycerol) and lysed in a Dounce tissue grinder with 20 strokes on ice. After lysis, crude extracts were supplemented with 10 µl DNArase (c-Lecta), incubated 5 min on a tube roller mixer at 4 °C, and centrifuged for 25 min at 20,000 × $g$ at 4 °C. Cleared extracts were supplemented with 2 mg of avidin (E-proteins) and bound by gravity flow to 1 ml of StrepTactinXT 4Flow high capacity (IBA) equilibrated in Wash Buffer B (25 mM Tris pH 8.0, 300 mM NaCl, 2 mM β-mercaptoethanol + 5% glycerol). Resin was washed with 10 column volumes of Wash Buffer B and eluted with Elution Buffer B (Wash Buffer B + 50 mM biotin). Peak fractions containing BICD2 were identified by SDS-PAGE, pooled, concentrated to 300–350 µl using Vivaspin 6 30,000 MWCO concentrators (Sartorius), and loaded onto a Superdex 200 10/300 GL column equilibrated in SEC Buffer (25 mM Tris pH 8.0, 300 mM NaCl, 0.5 mM TCEP + 5% glycerol). Peak fractions were pooled, concentrated with Vivaspin 500 30,000 MWCO concentrators to 0.9–2 mg/ml, aliquoted, snap-frozen in liquid nitrogen and stored at −80 °C until further use.

## Protein phosphorylation and phosphosite determination
In vitro protein kinase assay was carried out by incubation of 1 µg purified protein substrates in the presence or absence of 100 ng PLK1 (Thermo Fisher) or CDK1/cyclin B1 (Thermo Fisher) in Phosphorylation Buffer (50 mM MOPS pH 7.4, 5 mM MgCl₂, 10 mM β-glycerophosphate, 1 mM EGTA, 1 mM DTT plus 100 µM ATP) at 25 °C for the indicated times. Reactions were stopped with electrophoresis sample buffer and after SDS−PAGE proteins were visualized by Coomassie Brilliant Blue staining. Phosphate incorporation was visualized by adding 1 µCi $^{32}$P-γ-ATP (200 cpm/pmol, Perkin Elmer), plus autoradiography or a PhosphorImager system (Molecular Dynamics). Quantifications were done using the PhosphorImager system. Non-radioactive reaction duplicates were used for phosphosite determination. After SDS-PAGE, relevant bands were cut, digested with trypsin or, when indicated, chymotrypsin and analyzed by nano-LC-MS/MS using an Orbitrap Fusion Lumos TM Tribrid instrument at the IRB Barcelona Mass Spectrometry and Proteomics Core Facility. Samples were run through a PepMap100 C18 u-precolumn (Thermo Fisher) and then separated using a C18 analytical column (Acclaim PepMap RSLC, Thermo Fisher) with a 90 min run, comprising three consecutive steps with linear gradients from 3 to 35% B in 60 min, from 35 to 50% B in 5 min, and from 50% to 85% B in 1 min, followed by isocratic elution at 85 % B in 5 min and stabilization to initial conditions (A = 0.1% FA in water, B = 0.1% FA in CH₃CN). The column outlet was directly connected to an Advion TriVersa NanoMate (Advion) fitted on an Orbitrap Fusion Lumos™ Tribrid (Thermo Scientific). The mass spectrometer was operated in a data-dependent acquisition (DDA) mode. Survey MS scans were acquired in the orbitrap with the resolution (defined at 200 $m/z$) set to 120,000. The lock mass was user-defined at 445.12 $m/z$ in each Orbitrap scan. The top speed (most intense) ions per scan were fragmented in the HCD cell and detected in the orbitrap. The ion count target value was 400,000 and 50,000 for the survey scan and for the MS/MS scan respectively. Target ions already selected for MS/MS were dynamically excluded for 30 s. Spray voltage in the NanoMate source was set to 1.60 kV. RF Lens were tuned to 30%. Minimal signal required to trigger MS to MS/MS switch was set to 25,000. The spectrometer was working in positive polarity mode and singly charge state precursors were rejected for fragmentation. A database search was performed with Proteome Discoverer software v2.1 (Thermo Scientific) using Sequest HT, Amanda search engines and SwissProt database Mouse release 2016_11; contaminants database and user proteins manually introduced. Searches were run against targeted and decoy databases to determine the false discovery rate (FDR). Search parameters included trypsin enzyme specificity, allowing for two missed cleavage sites, Methionine oxidation, Phosphorylation in serine/threonine/tyrosine and acetylation in N-terminal as dynamic modifications and Carbamidomethyl in cysteine as static modification. Peptide mass tolerance was 10 ppm and the MS/MS tolerance was 0.02 Da. Peptides with a $q$-value lower than 0.1 and an FDR < 1% were considered as positive identifications with a high confidence level. The PhosphoRS node was used to provide a confidence measure for the localization of phosphorylation in the peptide sequences identified with this modification.

## Cell culture, synchronization and transfection
HeLa (CCL-2, obtained from ATCC) and U2OS (HTB-96, obtained from ATCC) cells were grown in DMEM media (Thermo Fisher) supplemented with 10% foetal bovine serum (Thermo Fisher), 2mM L-glutamine (Thermo Fisher) and 100 U/mL penicillin/ 100 µg/mL streptomycin (Thermo Fisher).

Mitotic HeLa cells were obtained by mitotic shake off of cells arrested using 200 ng/mL nocodazole (Merck Life Science) for 16 h. When indicated 100 nM BI2563 (Adooq Bioscience) or 5 µM STLC (Merck Life Science) was used for 16 h. Cells arrested in G2 were obtained after a 16 h treatment with 9 µM RO-3306 (Enzo Life Sciences). Cells at the G1/S border and progressing through G2 were obtained using a double thymidine double block. Briefly, cells were plated with media containing 2 mM thymidine (Merck Life Science) during 16 h, washed and released in fresh media. After 8 h post-release media was supplemented again with 2 mM thymidine during 16 h. Cells collected at this point were in G1/S. To obtain G2 cells, after the double thymidine block cells were washed, released in fresh media and collected after 8 h. Cell cycle phase was confirmed by FACS in all cases (see Supplementary Fig. S7A).

For immunoprecipitation experiments, cells were transfected using linear polyethylenimine (Polysciences Inc.) according to manufacturer's instructions. For immunofluorescence, either Lipofectamine 2000 (plasmids) or Lipofectamine RNAiMAX (siRNAs) was used, according to manufacturer's instructions (Thermo Fisher).

## Antibodies
Primary antibodies and dilutions used were as follows:

α-PLK1 (mouse monoclonal IgG2b, Calbiochem #DR1037; WB, 1:1000).

α-BICD2 (rabbit polyclonal, Abcam #ab117818 for western blot (1:1000) and immunoprecipitations (IP); for immunofluorescence (1:300), rabbit polyclonal #2293, a gift from Anna Akhmanova (Utrecht University), described in ref. 14).

α-GFP (mouse monoclonal IgG2a, Thermo Fisher #A11120; WB, 1:1000; IF, 1:1000); rabbit polyclonal Torrey Pines #TP401 (IF: 1:500) and Santa Cruz #sc-8334 (IF, 1:500).

α-p150 (mouse monoclonal IgG2b, Santa Cruz #sc-365274; WB, 1:1000).

α-DIC (mouse monoclonal IgG2b, Santa Cruz #sc-13524 for WB (1:1000) and IP; mouse monoclonal IgG2b, Thermo Fisher #14- 97772-80 for IF(1:500)).

α-GST (mouse monoclonal IgG1, Sigma #SAB4200237; WB: 1:1000).

α-GAPDH (mouse monoclonal IgG1, Santa Cruz #sc-47724; WB: 1:1000).

α-Tom20 (mouse monoclonal IgG2a, Santa Cruz #sc-17764; IF, 1:250).

α-PCNT (rabbit polyclonal, Abcam #ab4448; IF: 1:1000).

α-CycB (mouse monoclonal IgG1, Santa Cruz #sc-245; IF: 1:2000) Secondary antibodies used were as follows (Alexa Fluor conjugates, Thermo Fisher; HRP conjugates, R&D):

Alexa Fluor 647 goat anti-rabbit IgG (H + L) #A21244 (IF: 1:500–1:1000).

Alexa Fluor 568 goat anti-mouse IgG2b #A21144 (IF: 1:500–1:1000).

Alexa Fluor 488 goat anti-mouse IgG1 #A21121 (IF: 1:500–1:1000).

Alexa Fluor 488 goat anti-rabbit IgG (H + L) #A11008 (IF: 1:500–1:1000).

Alexa Fluor 647 goat anti-mouse IgG1 #A21240 (IF: 1:500–1:1000).

Alexa Fluor 488 goat anti-mouse IgG2a #A21131 (IF: 1:500–1:1000).

Alexa Fluor 568 goat anti-mouse IgG1 #A21124 (IF: 1:500–1:1000).

Alexa Fluor 647 goat anti-rabbit IgG (H + L) #A21244 (IF: 1:500–1:1000).

Goat α-mouse IgG-HRP #HAF007 (IF: 1:10,000).

Goat α-rabbit IgG-HRP #HAF008(IF: 1:10,000).

### Immunoprecipitation, pulldowns and western blot

Cells were lysed in Lysis Buffer B2 (20 mM Tris-HCl pH 8, 100 mM KCl, 1% Triton X-100, 10 mM β-glycerophosphate, 25 nM calyculin A, 0.5 mM PMSF, 1 µg/mL aprotinin and 1 µg/mL leupeptin), adapted from ref. 10. To study coimmunoprecipitation of BICD2 with dynein, we used Lysis Buffer D (25 mM HEPES pH 7.4, 50 mM potassium acetate, 2 mM magnesium acetate, 0.2% Triton X-100, 1 mM EGTA, 50 mM NaF, 10 mM β-glycerophosphate, 10% glycerol, 0.5 mM Mg-ATP, 1 mM DTT, 25 nM calyculin A, 0.5 mM PMSF, 1 µg/mL aprotinin and 1 µg/mL leupeptin), adapted from ref. 35. Immunoprecipitations and western blotting were performed using standard protocols as described in ref. 74 using the respective lysis buffers as wash buffer. For immunoprecipitation protein G coupled to Dynabeads (Thermo Fisher) was used. GST pulldowns were done as described in ref. 50 using glutathione agarose beads (GE Healthcare). In order to study the binding of BICD2 to GST-RanBP2 BBD and GST-Rab6, bacterial proteins attached to agarose beads were incubated for 90 min at RT with different forms of BICD2 purified from insect cells in Binding Buffer (50 mM Tris pH 7.4, 125 mM NaCl, 5 mM EDTA, 0.1% NP40). After that, beads were washed 4 times with Binding Buffer and resuspended in electrophoresis sample buffer. GST-Rab6 was preloaded with either GDP or GTPγS (200 nM in 25 mM Tris pH 7.5, 10 mM EGTA and 5 mM MgCl, 1 h at 25 °C). When specified, GST-RanBP2 BBD was pre-phosphorylated by CDK1/cyclin B1 (Thermo Fisher) in Phosphorylation Buffer plus ATP at 30 °C for 45 min.

### Immunofluorescence

Cells were grown on Poly-L-Lysine coated coverslips to subconfluence. After the indicated treatments, cells were rinsed in cold PBS and fixed with methanol at −20 °C for at least 15 min. After three washes in PBS, fixed cells were incubated for 20 min in Blocking Buffer (3% BSA, 0.1% Triton X-100, 0.02% NaN₃ in PBS) and stained using the indicated primary antibodies diluted in blocking solution for 1 h at room temperature. Secondary antibodies were diluted in blocking solution together with DAPI (0.01 mg/mL)

(Merck Life Science) and incubated for 20 min at room temperature in a dark, humid chamber. Slides were mounted in ProLong Gold Antifade Reagent mounting media (Thermo Fisher), sealed and stored in a cold dark chamber. To better visualize nuclear envelope proteins, nocodazole (10 µM) was included in the last 30 min prior to fixation[21].

G2 cells were identified by positive cyclin B1 cytoplasmic staining. Prophase cells were identified by apparent chromosome condensation as assessed with DAPI staining, intact nuclei as assessed by the shape of the DNA and the apparent exclusion of cytoplasmic markers from the nucleus, plus the appearance of nuclear cyclin B1 staining (independent experiments showed that these features paralleled histone H3[Ser10] phosphorylation in all cases, confirming correct cell cycle phase staging).

Three channel Z-stack images were acquired in a Leica AF6000 system with an Orca AG camera (Hamamatsu) coupled to a Leica DMI6000B microscope equipped with 63×/1.40NA and 100×/1.40NA oil immersion lens and the standard LAS-AF software followed with deconvolution. Four channel Z-stack images were acquired with a Leica Thunder system with a DMI8 microscope equipped with 100x NA 1.40 HCX PL-APO oil immersion lens and the standard LAS-AF software followed with deconvolution. Four channel Z-stack confocal images were obtained using a Zeiss Lsm780 confocal system with an inverted XYZ Zeiss Axio Observer Z1 microscope equipped with an 63×/1.40NA oil immersion lens and the standard Zeiss software (ZEN).

### Negative staining electronic microscope (EM)

Wild type and S102D BICD2 samples were diluted in Protein Buffer (25 mM Tris-HCl pH 9.0, 300 mM NaCl, 0.5 mM TCEP) to a final concentration of 0.02 mg/ml. Diluted samples were applied onto home-made glow-discharged carbon-coated copper grids and negatively stained with 2% (w/v) uranyl formate, pH 7.0. Electron microscope images were acquired in a FEI Tecnai G2 Spirit microscope with a Lab6 filament and operated at 120 kV, using a TVIPS CCD camera. Microscope images were processed using the cryoSPARC v3.2[75] software package. Automatically selected BICD2 particles (150,473 BICD2 wild-type particles and 106,650 BICD2 S102D particles) were aligned and classified using the reference-free 2D classification tool in cryoSPARC to obtain a clean dataset (30,476 BICD2 wild-type particles and 22,021 BICD2 S102D particles). Each image was classified into a sub-group and class average using unsupervised and reference-free image processing methods in cryoSPARC v3.2[75], therefore without any bias introduced by the user or from initial templates. After 2D classification, each subgroup was interpreted as corresponding to either the compact or partially extended conformation based on the aspect of the 2D average of each group. The number of particles for each conformation was estimated by adding the number of particles assigned to each of the 2D averages assigned to each conformation.

### Quantification and statistical analysis

Western blot quantification plus measurement of centrosome to nucleus and centrosome to centrosome distances were done using FIJI software. Assessment of mitochondria clustering and protein localization at the nuclear envelope was done using Leica LAS and FIJI software. Graphics and statistical analysis were done using Prism 9 software. Details of the number of measurements and biological replicates, as well of the statistical tests used in each case as well as P values are given in the corresponding figure legends (n.s. (not significant) $P > 0.05$; $*P < 0.05$; $**P < 0.01$; $***P < 0.001$).

### Reporting summary

Further information on research design is available in the Nature Portfolio Reporting Summary linked to this article.

## Data availability

The authors declare that all the data supporting the findings of this study are available within the article and its supplementary information files. Source data are provided with this paper.

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

## Acknowledgements

We are grateful to Anna Akhmanova (Utrecht University) for reagents and comments and Susana Eibes (Danish Cancer Society Research Center) for help with different DNA constructs and comments. We would also like to thank Elena Rebollo and the Molecular imaging Platform at the IBMB-CSIC as well as the Mass Spectrometry & Proteomics Core Facility at the IRB Barcelona for their help and input. J.R. and N.G.-S.work was funded by the Ministerio de Ciencia e Innovación (MICINN) and the Agencia Estatal de Investigación (AEI) from Spain (MCIN/ AEI /10.13039/ 501100011033/ FEDER "Una manera de hacer Europa") through Plan Nacional de I + D grants BFU2014-58422-P, PGC2018-096307-B-I00 and PID2021-127045NB-I00 (to J.R.) and by network grant 2017 SGR 1089 (AGAUR, Generalitat de Catalunya). N.G.-S. was a recipient of FPI Fellowship BES-2015-072446 from MICINN. Work in the laboratory of J.L. was supported by grants BFU2015-69275-P (MINECO/FEDER), PGC2018-099562-B-I00 (MICINN), network grants 2017 SGR 1089 (AGAUR) and RED2018-102723-T (MICINN), and by intramural funds of IRB Barcelona, supported by CERCA (Generalitat de Catalunya). F.Z. was supported by a fellowship from the "la Caixa" Foundation (ID 100010434, fellowship code LCF/BQ/DI17/11620020) and the European Union's Horizon 2020 research and innovation program under the Marie Skłodowska-Curie grant agreement no. 713673. J.P. was supported by a fellowship from MICINN (BES-2016-078003). O.L. and M.S. work was funded by the Agencia Estatal de Investigación (AEI/10.13039/501100011033), Ministerio de Ciencia e Innovación [PID2020-114429RB-I00] and by the Autonomous Region of Madrid, co-funded by the European Regional Development Fund (ERDF) [SAF2017-82632-P to O.L.]; Autonomous

Region of Madrid and co-funded by the European Social Fund and the European Regional Development Fund [Y2018/BIO4747 and P2018/NMT4443 to O.L.]; CNIO is supported by the National Institute of Health Carlos III. IRB Barcelona and CNIO are Severo Ochoa Award of Excellence recipients from MICINN.

## Author contributions

N.G-S. designed, performed and analyzed biochemical experiments, except BICD2 pulldowns, done by P.S-F-L. N.G.-S. and P.S-F-L. designed, performed and analyzed cell biology experiments, except mitochondria clustering assays that were performed by J.P. with the contribution of P.S-F-L. L.R. assisted with different cell and biochemical experiments and performed all experiments involving RanBP2. F.Z. expressed, purified and characterized recombinant BICD2 for structure analysis and interaction experiments. M.S. performed further characterization of BICD2, EM experiments and image analysis. O.L. and J.L. provided structural and cell biology expertise, respectively, as well as help designing the study, analyzing data and producing the manuscript. J.R. designed the study, analyzed the data and wrote the manuscript. All the authors reviewed the figures and manuscript.

## Competing interests

The authors declare no competing interests.
