## [Peer Review File · Nature Communications]

BICD2 phosphorylation regulates dynein function and centrosome separation in G2 and MREVIEWER COMMENTS

Reviewer #1 (Remarks to the Author):

General comments

BicD2 is a well-established regulatory factor for the motor protein cytoplasmic dynein. Although it is one of a class of such proteins, BicD2 is the most extensively studied. This paper reports the first direct phosphoregulatory mechanisms for BicD2. This is potentially a quite important advance, multiplying the number of BicD2-affected functions to include mitosis, and opening the door to a broad potential array of new dynein regulatory mechanisms.

BicD2 is known to regulate G2-dependent pre-mitotic nuclear transport behavior in non-neuronal cultured cells and neuronal progenitors in the developing brain. However, BicD2 has no known direct role in mitosis. The current results make an interesting case for such a function.

The results are mostly convincing, with the exception of the final data on centrosome behavior.

Specific comments.

1 - Nesprin-2G has been recently identified as a new nuclear BicD2 interactor, a point the authors should include in the text and, presumably their diagrams. It is uncertain which of the known BicD2 interactors might be involved in the newly described mitotic behavior, especially as there are no direct biochemical experiments in this study using RanBP2. These points should be discussed, but don't need to be addressed experimentally.

2 - Experiments showing an interaction between BicD2 and PLK1 are shown in Figure 1. The autoradiographs regarding BicD2 phosphorylation by PLK1 and CDK1 should also be in the main figures.

3- Two potential roles for BicD2 in centrosome behavior are considered, which seem to have become confused in the literature. One issue regards the nature of forces between nucleus and centrosome, and the other during centrosome separation. The first case seems to involve NE-associated dynein pulling on the centrosome-centered microtubules. The second case seems to involve repulsive kinesin (Eg5) generated forces. BicD2-dynein might, in addition, provide an additional repulsive force at this stage, though the authors should attempt to explain such a model in terms of MT organization and possible or known BicD2-dynein distribution. It is also uncertain what conditions are being compared in discussing the claim for an effect of BicD2 on centrosome separation. When N-C distances are shown in fig 6A and 6D there is a clear and significant effect of BicD2 RNAi compared to control, and this effect is rescued by expressing BicD2-GFP. But in figures 6B and 6E there is no effect of BicD2 RNAi compared to control, and, therefore, no effect to be rescued.

4 - In Fig. 2C the authors use WT and mutated versions of GFP-BicD2 (amino acids 1-575) fused to a mitochondria-targeting domain to measure mitochondrial clustering as a measure of dynein-driven motility in vivo. The authors should explain in greater detail the methodology used to measure clustering. One important missing condition is that for BicD2-S102A. The authors should test whether S102A decreases dynein motility.

5 - For western blots claimed to show a difference in binding affinity between the mutated versions of BicD2 and its interactors (dynein, PLK1 and CDK1) quantification of the bands, graphs of these quantifications and statistical analysis should also be shown.

6 - Is GFP expression the control for the GFP-BicD2 expression in the figures 5C, 5D, 6A, 6B, 6D and 6E? Authors should check as to whether the second column in the lane marked " GFP" the (-) sign should be (+). Same issue should be checked in Fig 6C, 6F and 5E.

7 – Finally, some of the most detailed information about G2- and potentially M- phase BicD2 function have come from the analysis of nuclear and neuronal migration in brain, especially in RGP cells. In particular, BicD2 was found to be important in nuclear migration in RGPs, but not directly in mitosis (see Doobin , D., et al., Nat. Commun. 2016). The relationship between that work and the current study should be discussed.

8 - Minor point. Format of reference (line 106)

Reviewer #2 (Remarks to the Author):

The authors study the regulation of the dynein adaptor BicD2 by Plk1 kinase in G2/M. They identify Ser102 of BicD2 as a major Plk1 phosphorylation site and show using single point mutants in cultured human cells evidence for phosphorylation at this site relieving BicD2 autoinhibition, controlling recruitment to the nuclear envelope and supporting centrosome separation. This work proposes a new cell cycle-dependent mechanism of BicD2 and hence dynein activity regulation. The manuscript is well written, the methods are detailed and clear, the claims are usually well supported by the data and the existing literature is appropriately cited. The discussion is a little long but provides interesting and relevant context.

Major concerns:

1. Evidence for BicD autoinhibition has previously been reported for *Drosophila* BicD. The authors do however not comment on whether Ser102 and the consensus sequence for Plk1 phosphorylation at this site is conserved in *Drosophila*. If it is not conserved in *Drosophila*, it would be appropriate to discuss which implications this has for the regulation of BicD2 in different organisms.

2. The cryo-EM data in Fig. 3 do not seem to be very convincing. A variety of different classes of more or less triangular or more or less linear BicD2 species is found for wildtype and the BicD2 mutant. These various different classes are then grouped into two major groups that are supposed to represent autoinhibited versus non-inhibited BicD2 forms. On which basis are the different class averages attributed to the two groups? In the simplest case, one would expect a non-inhibited BicD2 molecules (after opening up) to appear twice as long as the presumably autoinhibited molecules in which N and C-termini interact. However, triangular and linear structures seem to have roughly the same length, and both seem compatible with the N-terminal part interacting with the C-terminal part of the molecule. The linear structures appear to be collapsed triangular structures. How should we expect dynein/dynactin to bind? For a non-inhibited structure, one would expect to see an exposed part corresponding to the N-terminal part of BicD2 that does not interact with the rest of the molecule, but none of the structures appear to show such a conformation. The absence of a clear difference between the gel filtration elution profiles of wildtype and mutant BicD2 also seem to argue that there is no major conformational change of the mutant compared to wildtype. It seems therefore uncertain whether the cryo-EM images really show autoinhibited versus non-inhibited conformations.

3. The data in Fig. 4 are not particularly strong. For example the input band in the 5th lane of Fig. 5A is weaker than the other input bands, possibly accounting at least in part for a reduction of detected DIC in this condition. The only band providing evidence for DIC pulldown in Fig. 5B is rather weak. How often have these experiments been performed? The data could be presented in a more quantitative manner (average band intensities of different repeats of the experiment, normalising using the input intensities).

Minor points:

4. The Ser102 phosphorylation site seems to be far away from the polo box binding site. Is this distance comparable to other Plk1 substrates?

5. Presentation of pulldown western blots: For clarity, the expected molecular weights of the various fragments could be indicated in the figures or stated in the legends. Some blots are a little inaccessible:

(a) Fig. 1B: what are the various bands with different molecular weights detected by the anti-GFP antibody? Why is a band corresponding apparently to GFP alone detected in all lanes?

(b) Fig. 3C: Why is a band corresponding to the molecular weight of GST alone detected in all lanes?

6. Fig. 5: Can a more quantitative measure be provided for how the measured fluorescence intensities in panels A, C and D were assigned to the categories "negative", "weak/partial" and "high"?

7. Fig. 5: Panels E and F are not described in the text. Would it be more natural to first show raw data (B, E, F) and then the analysis (A, C, D)?

8. Fig. 6: It's unclear to this reviewer what we are supposed to see in the insets shown in panel C. More description/arrows to indicate points of interest in the images might be necessary to make this figure accessible to the non-specialist. How was the distance between centrosomes measured? Indicating this in the images would help. The outliers in the plots A, B, C cause the main clouds of data points to be very close to the x-axis. Using a y-axis with different scales for the bottom and top part might allow to visualize better the differences between the different conditions.

9. Typos: line 258: "fluoresce", line 464: "through"

Reviewer #3 (Remarks to the Author):

The authors of this manuscript set out to explore how dynein localization at the nuclear envelope is achieved. They looked towards the activating adaptor, BicD2, which functions to activate dynein during nuclear envelope breakdown and promotes dynein recruitment to the nucleus to drive nucleokinesis in migrating neural progenitor cells. They find that BicD2 is phosphorylated by CDK1 during M-phase. CDK1-phosphorylated BicD2 can then bind and be phosphorylated by PLK1 (at aa Ser102). The authors find PLK1 phosphorylation of Ser102 promotes association with dynein. They propose that the increase in association with dynein is because PLK phosphorylation of Ser102 promotes a conformation of BicD2 that is more "open" and less auto-inhibited. The authors also show that preventing Ser102 phosphorylation by PLK1 causes dynein and BicD2 nuclear envelope localization defects and impair nucleus-centrosome coupling and centrosome-centrosome coupling.

This work provides an important contribution to the field of dynein biology by revealing a novel way that BicD2 autoinhibition can be relieved. Pending the points below, I recommend this manuscript for publication.

1. All westerns should be repeated multiple times (and the "n" should be listed in the legend) and quantified in nearly every figure where conclusions were made from a western blot or an immunoprecipitation.

This is especially important in Figure 4 where the authors conclude that inhibiting both CDK1 and PLK1 inhibited dynein interaction. In this figure, it looks like the sample which was treated with RO and BL2h has less DIC in the input. By reporting the intensity of the anti-DIC signal in the IP to the intensity in the input, the authors could make a stronger case that inhibition of both kinases inhibits the association of BicD2 and dynein.

2. In some figures, I had a difficult time differentiating if BicD2 was migrating with a high MW (indicating that it was phosphorylated) (Example, in Figure 4 A, I had a hard time seeing the high MW BicD2 bands). Is the red arrowhead pointing to high MW BicD2 and the black arrowhead pointing to lower MW BicD2? Or are both arrowheads indicated higher MW BicD2 species, as the legend states? This point needs to be made clearer and should be illustrated with more clarity in the figure, as well.

3. Where is the SEC-MALS, DLS, and melting curve data for BicD2 WT vs S102D? These are important controls to show that mutation S102D does not change the gross behavior of BicD2. Because BicD2-S102D is used heavily in the rest of the paper to make substantive claims about BicD2 function, the authors should show this data in the supplemental portion of the manuscript.

4. The most surprising and interesting part of the paper was that preventing BicD2-S102 phosphorylation not only impaired dynein localization to the nucleus, but impaired BicD2 localization to the nucleus. In the discussion, the authors suggest that this means that PLK1 phosphorylation of BicD2 also regulated binding to RanBP2, which is BicD2's established nuclear binding partner. The authors have all the tools in place to test this directly (e.g. does BicD2 S102D bind to RanBP2 and S102A does not?). Determining if PLK1 phosphorylation and BicD2 opening also promotes RanBP2 binding would be a powerful addition to the manuscript that would increase its impact significantly since this quite different than how Rab6 activates BicD2 to bind dynein.

5. In many cases, I found the figure legends to contain insufficient information to interpret the figure without also following along closely in the text. This was especially true in the figure legend for Figure 6. More experimental details should be included in the legend for ease of reading.

6. A final model figure would help immensely and should be included.

7. How did the authors determine which 2D class averages in Figure 3A represented a triangular structure or a rod like structure? I found that some of the "rod" class averages for BicD2 S102D looked quite "triangular". (Look at the bottom most class average on the far left of the S102D sample, as well as

the class average in the top row that is second from the right). There is no experimental detail about how the researchers determined which class averages were rod-like, and which were triangular. If this was performed manually, the authors should blind the data analysis, since knowledge of sample identity could easily bias the rod vs triangular determination.

8. I found the cell images hard to see in Figure 6. All channels should be shown as non-merged, black and white (not false colored) images, in addition to showing the merged image. If space-allowance is limiting, the non-merged files can be placed in the supplemental portion of the manuscript

Reviewer #4 (Remarks to the Author):

Gallisa-Sune, Roig and colleagues here explore the phosphoregulatory mechanisms that control the dynein adaptor, BICD2, and impact on its localisation and control of dynein-regulated activities in G2 and mitosis. They show that phosphorylation of BICD2 facilitates its interaction with, and phosphorylation by, PLK1. This drives a conformational change within the molecule and allows it to localise dynein and ensure appropriate positioning of the centrosome as cells move into mitosis.

The experiments are clearly presented and appropriately controlled and the conclusions drawn by the authors are generally supported by their data. The description of the experimental methods would allow this work to be repeated. These findings advance our understanding of mitotic control processes through dynein-dynactin regulation.

I have only a few specific comments:

1. An experimental demonstration should be provided that CDK1-cyclin B is required *in vivo* for BICD2 phosphorylation. This is an important component of the authors' model, so that a *in vitro* assay should be supported by such data. It is notable that the dynein-BICD2 interaction is not wholly abrogated by CDK1 inhibition, so this conclusion may need to be tempered.

2. Figure 2C should include S102A data to demonstrate the need for phosphorylation to drive the active complex (the WT impact is still rather high).

3. Some indication of reproducibility should be provided; how many times were the immunoprecipitations or pulldowns done; quantitation should be provided for the immunoblots that

show qualitative changes, as distinct from binary results. Errors should be indicated for the in vitro phosphorylation experiments.

4. Although the Materials and Methods note that flow cytometry was used to verify the cell cycle arrests imposed by the drug treatments, these data should be included as confirmatory in the Supplemental data.

5. The immunoblots should be labelled with 'alpha' as an abbreviation for 'anti' throughout, rather than a-PLK1, etc.

6. The term 'G2/M' is not ideal. It would be more informative to specify 'G2 and M' phases, where the authors have performed specific inhibitory blocks to ensure that they are justified in making the distinction.

Response to the reviewers' comments.

We would like to thank all the reviewers for their time and comments, which definitely have helped us to improve the manuscript.

We have modified the original manuscript as specified below in the answers to the specific points raised by the reviewers. Please note that:

- We have added additional controls and quantifications to several experiments, as requested.
- As requested by reviewer #1, the original Figure 1 is now split in two (the new Figure 1 and 2) in order to move the data regarding the phosphorylation of BICD2 from the Supplementary Information section to the main figures. As a result of this, the numbering of all subsequent figures has changed from that of the original manuscript.
- We have added result using the pan-CDK inhibitor roscovitine in order to study the effect of inhibiting all CDKs on the interaction of dynein with BICD2 (Figure 5A).
- Following the suggestion from reviewer #3, an additional new set of results has been added, as Figure 8A. They show that the modification of BICD2 Ser102, in addition to favoring the interaction of the full-length adaptor with dynein and dynactin, greatly improves its interaction with the BICD2-binding domain (BBD) of RanBP2. We also show that the BBD needs to be previously phosphorylated by CDKs to facilitate this binding. This is unique for RanBP2 binding, as Rab6(GTP) can bind BICD2 independently of Ser102 modification (and CDK phosphorylation). We think that these results complete the description of the molecular mechanism regulating the binding of BICD2 to RanBP2 and ultimately of the recruitment of dynein to the nuclear envelope in G2 and early mitosis. They also explain some of our previous results (e.g. that BICD2 phosphonull mutants are not efficiently recruited to the nuclear envelope in G2). We discuss this in the new version of the manuscript.
- We now have a graphical model illustrating the proposed role of phosphorylation in the control of the activity and function of BICD2 in G2 and M (Figure 8B).

Additionally,

- We have added new data to the Supplementary Figures, thus also changing their original numbering. Importantly, we now have several new Supplementary Figures:
 - Figure S4, showing different biophysical analyses performed on purified BICD2.
 - Figure S5, showing deep learning-based predictions of the structure of BICD2.
 - Figure S7A, showing flow cytometry data that verify the cell cycle arrests imposed by the drug treatments in Figure 5.
 - Figure S9, showing the different channels of the images shown in Figure 7.
- Additionally, we have moved the original Figure 4D to Supplementary Materials (now figure S6), as we felt that the actual Figure 4C conveys the result of this set of experiments (i.e., that modification of Ser102 results in a weaker intramolecular interaction between the N- and C-terminal regions of BICD2) strongly enough. Moreover, this allowed us to add the quantification of Figure 4C without using extra space in the main figures.

Reviewer #1 (Remarks to the Author):

General comments

BicD2 is a well-established regulatory factor for the motor protein cytoplasmic dynein. Although it is one of a class of such proteins, BicD2 is the most extensively studied. This paper reports the first direct phosphoregulatory mechanisms for BicD2. This is potentially a quite important advance, multiplying the number of BicD2-affected functions to include mitosis, and opening the door to a broad potential array of new dynein regulatory mechanisms.

BicD2 is known to regulate G2-dependent pre-mitotic nuclear transport behavior in non-neuronal cultured cells and neuronal progenitors in the developing brain. However, BicD2 has no known direct role in mitosis. The current results make an interesting case for such a function.

The results are mostly convincing, with the exception of the final data on centrosome behavior.

We thank the reviewer for all the comments. We answer them below.

Specific comments.

1 - Nesprin-2G has been recently identified as a new nuclear BicD2 interactor, a point the authors should include in the text and, presumably their diagrams. It is uncertain which of the known BicD2 interactors might be involved in the newly described mitotic behavior, especially as there are no direct biochemical experiments in this study using RanBP2. These points should be discussed, but don't need to be addressed experimentally.

We have now added an additional experiment to our manuscript (Figure 8A) that shows that phosphorylation greatly favors the interaction of BICD2 with RanBP2. This is dependent on CDK1 phosphorylation of the BICD2 Binding Domain (BBD) of RanBPs. Our results thus suggest that the nucleoporin may be a major receptor for modified BICD2 in G2/M. Nevertheless, we agree that Nesprin-2 could certainly also play a part in the recruitment of phosphorylated BICD2 to the nuclear envelope in G2. Although we have not specifically studied this, we now mention it in the introduction, the discussion as well as in the figure legend of Figure 8B corresponding to the graphical model of our results.

2 - Experiments showing an interaction between BicD2 and PLK1 are shown in Figure 1. The autoradiographs regarding BicD2 phosphorylation by PLK1 and CDK1 should also be in the main figures.

We have now moved the autoradiographs to the main figures. For this we have reorganized them, showing the initial experiments describing the binding of PLK1 to BICD2 in Figure 1 and experiments regarding phosphorylation in Figure 2.

3- Two potential roles for BicD2 in centrosome behavior are considered, which seem to have become confused in the literature. One issue regards the nature of forces between nucleus and centrosome, and the other during centrosome separation. The first case seems to involve

NE-associated dynein pulling on the centrosome-centered microtubules. The second case seems to involve repulsive kinesin (Eg5) generated forces. BicD2-dynein might, in addition, provide an additional repulsive force at this stage, though the authors should attempt to explain such a model in terms of MT organization and possible or known BicD2-dynein distribution. It is also uncertain what conditions are being compared in discussing the claim for an effect of BicD2 on centrosome separation. When N-C distances are shown in fig 6A and 6D there is a clear and significant effect of BicD2 RNAi compared to control, and this effect is rescued by expressing BicD2-GFP. But in figures 6B and 6E there is no effect of BicD2 RNAi compared to control, and, therefore, no effect to be rescued.

We agree that BICD2-dynein most probably has two different yet related roles controlling centrosome positioning in G2 and M. Namely, tethering centrosomes to the nucleus and contributing to the forces that physically separate the organelles.

- In view of our data as well as previously published reports, in G2 HeLa cells the adaptor and the motor uniquely control centrosome attachment to the nuclear envelope, as in this cell line centrosome separation occurs later, in early M (this may not be the case for other cell types, for example, in MEFs we find that centrosomes separate in G2, see Eibes *et al. Curr Biol* **28**:121-129.e4. doi:[10.1016/j.cub.2017.11.046](https://doi.org/10.1016/j.cub.2017.11.046)). Accordingly, we do not observe any effect on inter-centrosome distances upon BICD2 RNAi (Figure 7C, previously Figure 6B) as no forces seem to exist at that point that can successfully push centrosomes apart even when they are detached from the nucleus. In this regard, note that in HeLa cells Eg5 is still not concentrated around centrosomes in G2 and we have previously found that the downregulation of the kinesin has no effect on intercentrosomal distance in this phase of the cell cycle (see Bertran *et al. EMBO J* **30**:2634–47. doi:[10.1038/emboj.2011.179](https://doi.org/10.1038/emboj.2011.179)).

- The situation in early mitosis is different. Here BICD2 and dynein are still controlling the tethering of the centrosomes to the nucleus and thus in part their position. But forces exist (e.g. resulting from the action of Eg5 plus the pushing resulting from the growing of the two asters) that drive centrosomes apart. In addition, the intercentrosomal linkage has been disassembled, and this allows the organelles to move apart more easily. We think that this explains our observations in BICD2-deficient prophase cells (Figure 7F, previously 6E): inter-centrosome distances are large (and similar to controls) as the result of centrosomes being detached from the NE and drifting apart as a result of a lack of connection to the NE and to one another or of the action of motors such as Eg5. We agree with the reviewer in that BICD2-dynein might provide centrosome to centrosome repulsive forces in early mitosis (as has been shown by Raaijmakers *et al. EMBO J* **31**:4179–90. doi:[10.1038/emboj.2012.272](https://doi.org/10.1038/emboj.2012.272)), but these forces do not seem to be necessary for centrosome separation in this phase in the context of normal Eg5 levels.

In any case, the relationship between centrosome position in early mitosis and the action of dynein (bound to BICD2 or possibly alternative adaptors) and other motors such as Eg5 and kinesin-1 is most probably complex. In fact, dynein has been shown to participate in centrosome separation in *Drosophila* and *C.elegans*, but also reported to be dispensable for the separation of centrosomes in prophase in U2OS and RPE-1 cells, or even to counteract Eg5 in early M (e.g. Tanenbaum *et al. EMBO J* **27**:3235–45. doi:[10.1038/emboj.2008.242](https://doi.org/10.1038/emboj.2008.242)). The complex roles of dynein during centrosome separation may somehow explain the dominant negative effect that our BICD2 phosphonull mutants have on centrosome separation in prophase (original Figure 6E, now 7F). Further experiments, e.g. characterizing the

relationship between Eg5 and BICD2-dynein, something that we plan to tackle in the future, will be needed to characterize this in detail, although we feel that this falls outside the scope of this report.

We have sought to better clarify these points, better separating dynein's two independent roles during centrosome separation, in the new version of the manuscript.

4 - In Fig. 2C the authors use WT and mutated versions of GFP-BicD2 (amino acids 1-575) fused to a mitochondria-targeting domain to measure mitochondrial clustering as a measure of dynein-driven motility in vivo. The authors should explain in greater detail the methodology used to measure clustering. One important missing condition is that for BicD2-S102A. The authors should test whether S102A decreases dynein motility.

We have now added data with BICD2 S102A to figure 2C (now 3D) and expanded our description of the methodology used. We find that, although BICD2 S102A tends to induce slightly less mitochondria clustering than wild type BICD2, the difference is only marginally significant ($p=0.0464$). We interpret this data as indicating that phospho-independent forms of BICD2 activation have a significant role in our assay, which in fact deals with interphase cells. An alternative or additional possibility is that the mitochondria targeting domain at the C-terminus of BICD2 is able to partially induce the formation of active dynein complexes (note also the behavior of wt BICD2). That said, we think it is important to note that the phosphomimetic S102D mutant has a clear and significantly bigger dynein-activating activity than both the wild type and S102A forms of BICD2, strongly suggesting together with the rest of our data that the modification of Ser102 favors BICD2 binding to dynein and the activation of the resulting motor complex.

5 - For western blots claimed to show a difference in binding affinity between the mutated versions of BicD2 and its interactors (dynein, PLK1 and CDK1) quantification of the bands, graphs of these quantifications and statistical analysis should also be shown.

We now show quantifications plus statistical analysis of all the western blots from which we draw quantitative conclusions. We also provide source data of all the experiments used to perform said analysis.

6 - Is GFP expression the control for the GFP-BicD2 expression in the figures 5C, 5D, 6A, 6B, 6D and 6E? Authors should check as to whether the second column in the lane marked "GFP" the (-) sign should be (+). Same issue should be checked in Fig 6C, 6F and 5E.

Indeed, GFP is the protein expressed as a control in these experiments. We acknowledge that our labeling may have been confusing and we have changed it in these and other figures.

7 - Finally, some of the most detailed information about G2- and potentially M- phase BicD2 function have come from the analysis of nuclear and neuronal migration in brain, especially in

RGP cells. In particular, BicD2 was found to be important in nuclear migration in RGPs, but not directly in mitosis (see Doobin , D., et al., Nat. Commun. 2016). The relationship between that work and the current study should be discussed.

We now give more emphasis to the importance of BICD2 in RPG nuclear migration both in the introduction and the discussion. Furthermore, we discuss the relationship between our results and this process.

8 - Minor point. Format of reference (line 106)

This has now been corrected. We thank the reviewer for noting this and, once more, for all his/her comments.

Reviewer #2 (Remarks to the Author):

The authors study the regulation of the dynein adaptor BicD2 by Plk1 kinase in G2/M. They identify Ser102 of BicD2 as a major Plk1 phosphorylation site and show using single point mutants in cultured human cells evidence for phosphorylation at this site relieving BicD2 autoinhibition, controlling recruitment to the nuclear envelope and supporting centrosome separation. This work proposes a new cell cycle-dependent mechanism of BicD2 and hence dynein activity regulation. The manuscript is well written, the methods are detailed and clear, the claims are usually well supported by the data and the existing literature is appropriately cited. The discussion is a little long but provides interesting and relevant context.

We thank the reviewer for the comments. We answer them below.

Major concerns:

1. Evidence for BicD autoinhibition has previously been reported for *Drosophila* BicD. The authors do however not comment on whether Ser102 and the consensus sequence for Plk1 phosphorylation at this site is conserved in *Drosophila*. If it is not conserved in *Drosophila*, it would be appropriate to discuss which implications this has for the regulation of BicD2 in different organisms.

Indeed, Ser102 is not conserved in *Drosophila* (in contrast to what we observe in for example *C. elegans*). This may mean that BICD lacks the regulatory input described in our work. Nevertheless, a putative PLK1 phosphorylation site (ETS₁₀₉L, that conforms to the PLK1 consensus motif [D/E]X[S/T]Φ, where Φ is an hydrophobic residue) is present 11 residues towards the C-terminus of the protein. We favor the hypothesis that BICD Ser109 (not

conserved in mammalian BICD2), is functionally equivalent to human BICD2 Ser102 and thus that the regulation of BICD family members by Polo-family kinases has been conserved through evolution. We have modified figure S2 to show this, and commented on it in the discussion.

(Incidentally, no clear “alternative” PLK1 site similar to that of *Drosophila* BICD seems to be present in the N-terminus of the BICD2 paralogues BICL1 or BICL2, suggesting that these proteins are indeed not subject to PLK1 regulation.)

2. The cryo-EM data in Fig. 3 do not seem to be very convincing. A variety of different classes of more or less triangular or more or less linear BicD2 species is found for wildtype and the BicD2 mutant. These various different classes are then grouped into two major groups that are supposed to represent autoinhibited versus non-inhibited BicD2 forms. On which basis are the different class averages attributed to the two groups?

EM images were classified into several subgroups using unsupervised and reference-free image classification and 2D averaging, which is user unbiased and the number of images for each conformation was determined by counting the images assigned to each group. The only user intervention is when assigning each average to a compact or a partially extended conformation, which is based on the visual inspection of the final 2D average for each subgroup. We have clarified this in the Results section and also in Methods.

In the simplest case, one would expect a non-inhibited BicD2 molecules (after opening up) to appear twice as long as the presumably autoinhibited molecules in which N and C-termini interact. However, triangular and linear structures seem to have roughly the same length, and both seem compatible with the N-terminal part interacting with the C-terminal part of the molecule. The linear structures appear to be collapsed triangular structures. How should we expect dynein/dynactin to bind? For a non-inhibited structure, one would expect to see an exposed part corresponding to the N-terminal part of BicD2 that does not interact with the rest of the molecule, but none of the structures appear to show such a conformation. The absence of a clear difference between the gel filtration elution profiles of wildtype and mutant BicD2 also seem to argue that there is no major conformational change of the mutant compared to wildtype. It seems therefore uncertain whether the cryo-EM images really show autoinhibited versus non-inhibited conformations.

We agree with the reviewer that our results from the EM experiments were not well explained in the previous version of our manuscript.

We have now addressed this issue in full. For this:

- The corresponding section in the Results part has been rewritten and clarified.
- We have made extended predictive analysis using AlphaFold to help in the interpretation of the EM images.

- We have made use of the recent EM analysis of BicD2, where they describe that the protein can transit between a compact, partially extended and fully extended conformation (Fagiewicz, R. et al. In vitro characterization of the full-length human dynein-1 cargo adaptor BicD2. *Struct. Lond. Engl.* 1993 S0969-2126(22)00353-7 (2022) doi:10.1016/j.str.2022.08.009.).

After all these changes, we now indicate in the manuscript that:

- The triangularly-shaped images are similar to the compact conformation described recently for BICD2 using EM by Fagiewicz, *et al.* where coiled-coils are not placed one after the other and as a result do not occupy their maximum possible length.
- Rod-like particles were similar to the partially extended conformation described for BICD2 by Fagiewicz, *et al.*.
- The S102D mutation does not induce a fully extended conformation, which would result in much longer molecules in the microscope, as mentioned by the reviewer. This agrees with our results using several biophysical techniques that indicate that the S102D mutation does not cause the large conformational changes expected for the fully extended conformation.

This could be interpreted as indicating that the mutation facilitates the transition of BICD2 from a compact to a partially extended conformation.

- In addition, the structural models for BicD2 obtained using Alphafold predict that the C-terminal CC3 could form a head to tail interaction with the N-terminal CC1. Residue S102 is located in a proximal position of the place of this intramolecular interaction.

We believe that after all these changes we have addressed in full the issue raised by the reviewer concerning the EM experiments.

In addition, we include a paragraph in the Results section to indicate that although our favored interpretation is that the S102D mutation helps the transition from a compact to a partially extended conformation, we cannot fully rule out that the triangularly-shaped and rod-shaped images could just represent a different view of BICD2 imaged in the microscope from a different angle, and that the S102D mutation affects the ratio of each view of the molecule detected in EM after the interaction with the carbon-coated support film. Even in this scenario, the differences observed in EM between the wild type and the mutant can only be a consequence of modifications, even if subtle, in the structure of the protein caused by the phosphomimetic mutation.

3. The data in Fig. 4 are not particularly strong. For example the input band in the 5th lane of Fig. 5A is weaker than the other input bands, possibly accounting at least in part for a reduction of detected DIC in this condition. The only band providing evidence for DIC pulldown in Fig. 5B is rather weak. How often have these experiments been performed? The data could be presented in a more quantitative manner (average band intensities of different repeats of the experiment, normalising using the input intensities).

For these and other figures we now show quantifications plus statistical analysis of different experiments. We note in the figure legends how many times the experiment has been performed. We also provide source data of all the experiments used to perform said analysis (e.g., 3 experiments for both figure 5A and B).

Minor points:

4. The Ser102 phosphorylation site seems to be far away from the polo box binding site. Is this distance comparable to other Plk1 substrates?

In a well-studied substrate such as Cdc25C the distance seems to be less, at least for some of the modified sites. e.g., PLK1 PBD binding around residue 130 (see Elia *et al.* Science 299:1228–31. doi:10.1126/science.1079079) and phosphorylating Ser198 (Toyoshima-Morimoto *et al.* EMBO Rep 3:341–348; see also

<https://www.phosphosite.org/proteinAction.action?id=938>). But in other cases binding and phosphorylation sites can be far apart. E.g., in NEK9, a protein kinase which we have studied in depth, PLK1 PBD binds in the vicinity of Ser869 and phosphorylates among others Thr210 (Bertran *et al.* EMBO J 30:2634–47. doi:10.1038/emboj.2011.179).

Probably the three-dimensional structure of the protein and its flexibility dictates the maximum distance between the PLK1 binding site and the sites that it can phosphorylate. For BICD2 we can hypothesize that a “closed” conformation can facilitate it (see figure S5). Although the N-terminus is most probably an extended rod, the PBD binding site (around residues 219-321) lies in the unstructured region in between CC1 and CC2 and the folding of this region towards the N-terminus may facilitate PLK1 phosphorylation of Ser102.

5. Presentation of pulldown western blots: For clarity, the expected molecular weights of the various fragments could be indicated in the figures or stated in the legends. Some blots are a little inaccessible:

(a) Fig. 1B: what are the various bands with different molecular weights detected by the anti-GFP antibody? Why is a band corresponding apparently to GFP alone detected in all lanes?

(b) Fig. 3C: Why is a band corresponding to the molecular weight of GST alone detected in all lanes?

We sometimes find that different fragments of GFP-BICD2 are produced when the recombinant protein is expressed in cells. This includes what seems to be GFP alone and may be the result of proteolysis in the linker between the fluorescent protein and BICD2. The effect is more visible in the immunoprecipitates. We have relabeled Figure 1B seeking to clarify it, also adding the expected MW of the fragments expressed to the figure legends.

Something similar can be observed for GST-tagged BICD2 (as can be seen in what is now Figure 4C, previously 3C). And it is also observable for GST-BBD expressed in bacteria (new figure 8A). In both experiments GST alone is observed in addition to the corresponding GST-tagged fragments.

In all cases, we have sought to clarify this in the Figure Legends.

6. Fig. 5: Can a more quantitative measure be provided for how the measured fluorescence intensities in panels A, C and D were assigned to the categories "negative", "weak/partial" and "high"?

We have tried alternative ways to assess protein accumulation and intensities at the nuclear envelope (e.g. by using intensity profiles of lines drawn across the nuclei). We find that as the intensities of the different proteins are very variable between specific regions around the perimeter of the nucleus, a categorization of cells as the one we have used is the most reliable manner to convey the results.

In this we follow previous literature in the field, such as Hu *et al.* Cell 154:1300–13. doi:10.1016/j.cell.2013.08.024 and Baffet *et al.* Dev Cell 33:703–16. doi:10.1016/j.devcel.2015.04.022. These studies categorize the cells as being positive or negative for different proteins (i.e. dynein) at the nuclear envelope in an equivalent semi quantitative manner as the one we use. In our manuscript we have added an additional category, "weak/partial", as we think that in this manner we convey in a more nuanced way the observed result.

7. Fig. 5: Panels E and F are not described in the text. Would it be more natural to first show raw data (B, E, F) and then the analysis (A, C, D)?

Agreeing with the reviewer that it may be more natural to first show the raw data, we have reorganized this and the next figure accordingly (new Figures 6 and 7). We now also mention the original E and F panels in the text (now Figure 6C and D).

8. Fig. 6: It's unclear to this reviewer what we are supposed to see in the insets shown in panel C. More description/arrows to indicate points of interest in the images might be necessary to make this figure accessible to the non-specialist. How was the distance between centrosomes measured? Indicating this in the images would help. The outliers in the plots A, B, C cause the main clouds of data points to be very close to the x-axis. Using a y-axis with different scales for the bottom and top part might allow to visualize better the differences between the different conditions.

Seeking to make the figure clearer we have made the following modifications:

- We now describe the significance of the insets in the figure legend (of now Figure 7). Insets show that cells are positive for cyclin B (red channel) and express GFP-tagged recombinant proteins (green channel).
- We explain better how we identify G2 and prophase cells.
- We have added schematics that seek to clarify the different distances quantified.
- Following the reviewer's suggestion, we have modified the graph axis, thus better visualizing the differences between conditions.

Also note, that as requested by reviewer #3, we now show the individual non-merged channels of Figure 7 in a new Supplementary Figure S9.

9. Typos: line 258: "fluoresce", line 464: "through"

This has now been corrected. We thank the reviewer for noting this and for all his/her comments.

Reviewer #3 (Remarks to the Author):

The authors of this manuscript set out to explore how dynein localization at the nuclear envelope is achieved. They looked towards the activating adaptor, BicD2, which functions to activate dynein during nuclear envelope breakdown and promotes dynein recruitment to the nucleus to drive nucleokinesis in migrating neural progenitor cells. They find that BicD2 is phosphorylated by CDK1 during M-phase. CDK1-phosphorylated BicD2 can then bind and be phosphorylated by PLK1 (at aa Ser102). The authors find PLK1 phosphorylation of Ser102 promotes association with dynein. They propose that the increase in association with dynein is because PLK phosphorylation of Ser102 promotes a conformation of BicD2 that is more "open" and less auto-inhibited. The authors also show that preventing Ser102 phosphorylation by PLK1 causes dynein and BicD2 nuclear envelope localization defects and impair nucleus-centrosome coupling and centrosome-centrosome coupling.

This work provides an important contribution to the field of dynein biology by revealing a novel way that BicD2 autoinhibition can be relieved. Pending the points below, I recommend this manuscript for publication.

We thank the reviewer for the comments. We answer them below.

1. All westerns should be repeated multiple times (and the "n" should be listed in the legend) and quantified in nearly every figure where conclusions were made from a western blot or an immunoprecipitation.

This is especially important in Figure 4 where the authors conclude that inhibiting both CDK1 and PLK1 inhibited dynein interaction. In this figure, it looks like the sample which was treated with RO and BL2h has less DIC in the input. By reporting the intensity of the anti-DIC signal in the IP to the intensity in the input, the authors could make a stronger case that inhibition of both kinases inhibits the association of BicD2 and dynein.

Answering to this and other comments from the reviewers, we now show quantifications plus statistical analysis of all the western blots in Figure 4 (now Figure 5) and other figures. Results are reported as the intensity of DIC signal normalized by the intensity of BICD2 signal in the

IPs as we think that this conveys better our results, we hope that the reviewer will agree. We also provide source data of all the experiments used to perform said analysis.

2. In some figures, I had a difficult time differentiating if BicD2 was migrating with a high MW (indicating that it was phosphorylated) (Example, in Figure 4 A, I had a hard time seeing the high MW BicD2 bands). Is the red arrowhead pointing to high MW BicD2 and the black arrowhead pointing to lower MW BicD2? Or are both arrowheads indicated higher MW BicD2 species, as the legend states? This point needs to be made clearer and should be illustrated with more clarity in the figure, as well.

In all figures the red arrowhead was meant to indicate the high apparent MW form of BICD2, and the black arrowhead the low apparent MW form. We now specify this in the figure legends of Figure 1 and 5 (previously Figure 4). Note that the MW shift is very apparent in mitotic (vs exponentially growing) cells, as can be seen in figure 1A and 5B, and not so easily observed (or not observed at all) in G2 cells (i.e. Figure 5A, where we have removed the arrowheads). We attribute this to a more complete modification of BICD2 once CDK1 is fully active.

3. Where is the SEC-MALS, DLS, and melting curve data for BicD2 WT vs S102D? These are important controls to show that mutation S102D does not change the gross behavior of BicD2. Because BicD2-S102D is used heavily in the rest of the paper to make substantive claims about BicD2 function, the authors should show this data in the supplemental portion of the manuscript.

We have added the requested data to the manuscript as Supplementary Figure S4 and referenced it in the text.

4. The most surprising and interesting part of the paper was that preventing BicD2-S102 phosphorylation not only impaired dynein localization to the nucleus, but impaired BicD2 localization to the nucleus. In the discussion, the authors suggest that this means that PLK1 phosphorylation of BicD2 also regulated binding to RanBP2, which is BicD2's established nuclear binding partner. The authors have all the tools in place to test this directly (e.g. does BicD2 S102D bind to RanBP2 and S102A does not?). Determining if PLK1 phosphorylation and BicD2 opening also promotes RanBP2 binding would be a powerful addition to the manuscript that would increase its impact significantly since this quite different than how Rab6 activates BicD2 to bind dynein.

We agree with the reviewer that the observation that Ser102 phosphorylation is necessary to recruit BICD2 to the nucleus is both surprising and interesting. In answer to the above comment, we have performed new experiments addressing whether the modification of Ser102 might regulate binding to RanBP2 (in addition to favoring the interaction with dynein and dynactin). In the new Figure 8A we show that, indeed, the modification of BICD2 Ser102 greatly increases the ability of full length BICD2 to bind to RanBPs BBD (BICD2 Binding Domain, residues 2147-2287) *in vitro*. In contrast, Rab6(GTP), although it may also bind BICD2 S102D a bit better than the wild type form, does not need Ser102 phosphorylation to

bind BICD2. Moreover, we show that the binding of BICD2 S102D to RanBP2 BBD is almost completely dependent of CDK1 phosphorylation of the BBD, an observation that fits well with previous reports (e.g. Baffet et al. *Dev Cell* 33:703–16. doi:10.1016/j.devcel.2015.04.022). Our new findings thus indeed suggest that phosphorylation regulates BICD2 in quite a different manner to that of Rab6.

5. In many cases, I found the figure legends to contain insufficient information to interpret the figure without also following along closely in the text. This was especially true in the figure legend for Figure 6. More experimental details should be included in the legend for ease of reading.

We have added more information to the legend for Figure 7 (originally Figure 6).

6. A final model figure would help immensely and should be included.

We now have added a model (Figure 8B) that schematizes our findings and seeks to illustrate the role of phosphorylation in controlling BICD2 activity and function in G2 and early mitosis.

7. How did the authors determine which 2D class averages in Figure 3A represented a triangular structure or a rod like structure? I found that some of the “rod” class averages for BicD2 S102D looked quite “triangular”. (Look at the bottom most class average on the far left of the S102D sample, as well as the class average in the top row that is second from the right). There is no experimental detail about how the researchers determined which class averages were rod-like, and which were triangular. If this was performed manually, the authors should blind the data analysis, since knowledge of sample identity could easily bias the rod vs triangular determination.

In the revised version, in the Results section and in Methods, we now indicate that EM images were classified into several subgroups using unsupervised and reference-free image classification and 2D averaging, which is user unbiased and the number of images for each conformation was determined by counting the images assigned to each group. The only user intervention is when assigning each average to a compact or a partially extended conformation, which is based on the visual inspection of the final 2D average for each subgroup.

We agree with the reviewer that a few “rod-like” 2D averages might look slightly “triangular”, probably reflecting that, as described by Fagiewicz et al (Fagiewicz, R. et al. *In vitro* characterization of the full-length human dynein-1 cargo adaptor BicD2. *Struct. Lond. Engl.* 1993 S0969-2126(22)00353–7 (2022) doi:10.1016/j.str.2022.08.009.), the protein can transit from fully compact to partially extended and fully extended conformations, and there is probably a continuum of conformations. However, this does not invalidate our results showing that there is a significant change in the ratio of images that can be ambiguously assigned to the “rod” or “triangular” conformation.

In addition, this potential uncertainty is now less significant in the context of the revised version of the manuscript. This section of Results has now been reformatted and clarified. We have

made an extended predictive analysis using Alphafold that helped us in the interpretation of the EM images. In addition, we have made use of the recent EM analysis of BicD2, where they describe that the protein can transit between a compact, partially extended and fully extended conformation (Fagiewicz, R. et al.). We conclude that the S102D mutation does not induce a fully extended conformation, which would reflect in much longer molecules in the microscope, but relatively minor conformational changes but which are however sufficient to promote a version of the protein that is more likely to form an active complex.

8. I found the cell images hard to see in Figure 6. All channels should be shown as non-merged, black and white (not false colored) images, in addition to showing the merged image. If space-allowance is limiting, the non-merged files can be placed in the supplemental portion of the manuscript

We have tried different ways of including the different image channels individually in the figure (that now also has lines indicating intra and intercentrosomal distances, possibly making them even more necessary). We think that with the space constraints of a single figure none of them results in a clear depiction of the results. We thus now show the individual non-merged channels in a new Supplementary Figure S9.

Once more, we thank the reviewer for all his/her comments.

Reviewer #4 (Remarks to the Author):

Gallisa-Sune, Roig and colleagues here explore the phosphoregulatory mechanisms that control the dynein adaptor, BICD2, and impact on its localisation and control of dynein-regulated activities in G2 and mitosis. They show that phosphorylation of BICD2 facilitates its interaction with, and phosphorylation by, PLK1. This drives a conformational change within the molecule and allows it to localise dynein and ensure appropriate positioning of the centrosome as cells move into mitosis.

The experiments are clearly presented and appropriately controlled and the conclusions drawn by the authors are generally supported by their data. The description of the experimental methods would allow this work to be repeated. These findings advance our understanding of mitotic control processes through dynein-dynactin regulation.

We would like to thank the reviewer for all the comments. We answer them below.

I have only a few specific comments:

1. An experimental demonstration should be provided that CDK1-cyclin B is required *in vivo* for BICD2 phosphorylation. This is an important component of the authors' model, so that a *in vitro* assay should be supported by such data. It is notable that the dynein-BICD2 interaction is not wholly abrogated by CDK1 inhibition, so this conclusion may need to be tempered.

Indeed, as can be seen in Figure 5A RO-3306 treatment and thus CDK1 inhibition does not completely abrogate the interaction between BICD2 and dynein *in vivo*. Concomitantly, RO-3306 does not completely interfere with the localization of BICD2 (and dynein) to the nuclear envelope.

In order to clarify this, we have used the pan-CDK inhibitor roscovitine to study the effect of inhibiting all CDKs in the interaction of dynein with BICD2 (Figure 5A). We now show that roscovitine completely abrogates the interaction of BICD2 with dynein *in vivo* (in a similar manner that BI2536 does) thus strongly suggesting that CDK phosphorylation of BICD2 is necessary for the interaction. This is supported with data showing that roscovitine strongly interferes with the localization of BICD2 and dynein to the nuclear envelope.

Altogether we interpret our data as suggesting that CDK1, together with different cyclins such as cyclin A in G2 and cyclin B in M, is indeed required *in vivo* for normal BICD2 phosphorylation and its interaction with dynein, but that in G2 other CDK family members (such as CDK2) can collaborate with it and partially compensate for its inhibition. We sought to convey this in the text.

2. Figure 2C should include S102A data to demonstrate the need for phosphorylation to drive the active complex (the WT impact is still rather high).

We now include the S102A mutant in the figure. As commented in the answers to reviewer #1, we find that BICD2 S102A tends to induce only slightly less mitochondria clustering than wild type BICD2. We propose that in this assay non-phospho-dependent mechanisms may have a significant role in the activation of BICD2, or possibly that the mitochondria targeting domain at the C-terminus of BICD2 is able to partially induce the formation of active dynein complexes.

In any case, the phosphomimetic S102D mutant has a clear and very significantly bigger dynein-activating activity, and we think that this strongly suggests together with the rest of our data that the modification of Ser102 favors BICD2 binding to dynein and the activation of the dynein motor complex.

3. Some indication of reproducibility should be provided; how many times were the immunoprecipitations or pulldowns done; quantitation should be provided for the immunoblots that show qualitative changes, as distinct from binary results. Errors should be indicated for the *in vitro* phosphorylation experiments.

We now indicate the number of times each experiment is done and provide quantifications and statistical analysis of the results. We also provide source data of all the experiments used to perform the analyses.

Regarding the *in vitro* phosphorylation experiments, we would like to note that in order to minimize the use of radioactivity we have performed the phosphorylation time courses only

once with [γ - ^{32}P]ATP. Nevertheless, in each case an additional parallel and independent "cold" version of the time course experiments was also performed. These experiments confirmed the phosphorylations detected by ^{32}P using MS. We think that this fact, plus the experimental design used, with its different time points (7 points per experiment), clearly help supporting our qualitative conclusions, i.e. that CDK1 mainly phosphorylates the C-terminus of BICD2 at $^{319}\text{TST}_{321}$ while PK1 prefers the N-t and Ser102 (Figures 2 and S1). With this in mind, in this only instance and noting that we do not make any quantitative claims from the results, we would like to ask the reviewer to leave the corresponding graphs as they are, without error bars.

4. Although the Materials and Methods note that flow cytometry was used to verify the cell cycle arrests imposed by the drug treatments, these data should be included as confirmatory in the Supplemental data.

We now show the data in Supplementary Figure S7A.

5. The immunoblots should be labelled with 'alpha' as an abbreviation for 'anti' throughout, rather than a-PLK1, etc.

We have changed this in all the figures as suggested.

6. The term 'G2/M' is not ideal. It would be more informative to specify 'G2 and M' phases, where the authors have performed specific inhibitory blocks to ensure that they are justified in making the distinction.

We have changed the title of the manuscript and the text accordingly and instead of using "G2/M" we now use "G2 and M" (followed by "phases of the cell cycle" when we think this could improve the clarity of the text).

We thank the reviewer once more for this and all the other comments.

REVIEWERS' COMMENTS

Reviewer #2 (Remarks to the Author):

The authors have carefully and satisfactorily addressed all my comments. Especially the description of the EM data is now clearer and more balanced. The additional quantifications of gel data do now also support the conclusions more clearly, and the reorganizations of the figures and editorial changes have overall improved clarity.

Reviewer #3 (Remarks to the Author):

The authors have done a great job addressing the comments of myself and Reviewer 1 and I support this revised manuscript for publication.

Reviewer #4 (Remarks to the Author):

The revision of this MS. has provided new depth and detail, as well as greatly clarifying elements of the model. I consider that the authors have addressed convincingly the substantive comments raised by the reviewers and that this study now presents an interesting advance in our knowledge of dynein-dynactin regulatory processes.

I have two points that the authors should address, although these are straightforward and might not necessitate further review:

1. It is not appropriate to derive standard deviations and statistical significance from only two repeats of an experiment, as in Figs. 1 and 2. Only the mean and the two values should be shown. Given the very clear differences seen in the experiments, I suggest that the two repeats are sufficient to validate the conclusions drawn by the authors.
2. The labelling within Fig. S4 should be revised. This may have been some sort of formatting/ pdf error, but the information within the various traces is unclear. Several of the axes are also not comprehensible.

RESPONSE TO THE REVIEWERS' COMMENTS

Once more, we would like to thank the reviewers for their comments.

As requested by Reviewer #4, for experiments that have only been repeated twice (corresponding to those in which the observed values for different condition were very clearly different) we now only show the obtained values and their mean.

The incorrections in Figure S4 were, as the reviewer suggested, the result of a pdf conversion error, and have now been fixed.

REVIEWERS' COMMENTS

Reviewer #2 (Remarks to the Author):

The authors have carefully and satisfactorily addressed all my comments. Especially the description of the EM data is now clearer and more balanced. The additional quantifications of gel data do now also support the conclusions more clearly, and the reorganizations of the figures and editorial changes have overall improved clarity.

Reviewer #3 (Remarks to the Author):

The authors have done a great job addressing the comments of myself and Reviewer 1 and I support this revised manuscript for publication.

Reviewer #4 (Remarks to the Author):

The revision of this MS. has provided new depth and detail, as well as greatly clarifying elements of the model. I consider that the authors have addressed convincingly the substantive comments raised by the reviewers and that this study now presents an interesting advance in our knowledge of dynein-dynactin regulatory processes.

I have two points that the authors should address, although these are straightforward and might not necessitate further review:

1. It is not appropriate to derive standard deviations and statistical significance from only two repeats of an experiment, as in Figs. 1 and 2. Only the mean and the two values should be shown. Given the very clear differences seen in the experiments, I suggest that the two repeats are sufficient to validate the conclusions drawn by the authors.
2. The labelling within Fig. S4 should be revised. This may have been some sort of formatting/ pdf error, but the information within the various traces is unclear. Several of the axes are also not comprehensible.